# Hyperpolyploidization of hepatocyte initiates preneoplastic lesion formation in the liver

Heng Lin[1,2,11], Yen-Sung Huang[3,4,11], Jean-Michel Fustin [5,6], Masao Doi [7], Huatao Chen [8,9], Hui-Huang Lai [10], Shu-Hui Lin[1,2], Yen-Lurk Lee[4], Pei-Chih King[1,2], Hsien-San Hou[4], Hao-Wen Chen[1,2], Pei-Yun Young[1,2] & Hsu-Wen Chao [1,2,11✉]

Hepatocellular carcinoma (HCC) is the most predominant primary malignancy in the liver. Genotoxic and genetic models have revealed that HCC cells are derived from hepatocytes, but where the critical region for tumor foci emergence is and how this transformation occurs are still unclear. Here, hyperpolyploidization of hepatocytes around the centrilobular (CL) region is demonstrated to be closely linked with the development of HCC cells after diethylnitrosamine treatment. We identify the CL region as a dominant lobule for accumulation of hyperpolyploid hepatocytes and preneoplastic tumor foci formation. We also demonstrate that upregulation of *Aurkb* plays a critical role in promoting hyperpolyploidization. Increase of AURKB phosphorylation is detected on the midbody during cytokinesis, causing abscission failure and hyperpolyploidization. Pharmacological inhibition of AURKB dramatically reduces nucleus size and tumor foci number surrounding the CL region in diethylnitrosamine-treated liver. Our work reveals an intimate molecular link between pathological hyperpolyploidy of CL hepatocytes and transformation into HCC cells.

[1] Department of Physiology, School of Medicine, College of Medicine, Taipei Medical University, Taipei 11031, Taiwan. [2] Graduate Institute of Medical Sciences, College of Medicine, Taipei Medical University, Taipei 11031, Taiwan. [3] The Ph.D. Program for Translational Medicine, College of Medical Science and Technology, Taipei Medical University, Taipei 11031, Taiwan. [4] Institute of Biomedical Sciences, Academia Sinica, Taipei 11529, Taiwan. [5] Laboratory of Molecular Metabolism, Graduate School of Pharmaceutical Sciences, Kyoto University, Sakyo-ku, Kyoto 606-8501, Japan. [6] The University of Manchester, Faculty of Biology, Medicine and Health, Oxford Road, Manchester M13 9PL, UK. [7] Department of Systems Biology, Graduate School of Pharmaceutical Sciences, Kyoto University, Sakyō-ku, Kyoto 606-8501, Japan. [8] Department of Clinical Veterinary Medicine, College of Veterinary Medicine, Northwest A&F University, Yangling, Shaanxi 712100, China. [9] Key Laboratory of Animal Biotechnology of the Ministry of Agriculture, Northwest A&F University, Yangling, Shaanxi 712100, China. [10] Department of Medical Laboratory Science and Biotechnology, College of Medicine, National Cheng Kung University, Tainan 70101, Taiwan. [11] These authors contributed equally: Heng Lin, Yen-Sung Huang, Hsu-Wen Chao. ✉email: chaohw3619@tmu.edu.tw

Hepatocellular carcinoma (HCC) is the most predominant primary malignancy in the liver, accounting for 90% of all liver cancer cases and is characterized by poor survival rate[1,2]. HCC develops progressively, after long-term exposure to carcinogenic agents causing random genetic variations[3]. Stemness traits, normally lacking from the adult liver, have been found in human HCC, raising the stem/progenitor cell origin hypothesis of liver cancers[4,5]. Recent findings, however, suggested that HCC is fairly heterogeneous and arises as a consequence of hepatocytes transformation during hepatocarcinogenesis[6,7]. Genetic lineage tracing approaches showed that HCC and hepatocellular adenoma (HCA) are derived from mature hepatocytes in mouse models[6,7]. Moreover, adult hepatocytes can trans-differentiate into biliary-like cells during liver cancer formation, and de-differentiate into progenitor-like cells in p53-deficient mouse liver[8,9]. The relevance of these models for human HCC is still a matter of debate. Despite increasing evidence on the origin of HCC cells, how hepatocytes transform into preneoplastic cells is still unclear.

Polyploidization has been proposed to play a critical role during tumorigenesis[10]. Most mammalian species are diploid, but polyploidy occurs in specific organs, such as in the liver during development, and has been suggested to signal the termination of differentiation or to enhance protection against stress conditions[11,12]. During postnatal development, hepatocytes progressively develop polyploidy[11,13] via incomplete cytokinesis without contractile ring formation, which is the major mechanism for generation of polyploid hepatocytes after weaning[11]. Recently, a population of hepatocytes failing abscission has been identified following the disruption of the *Period* genes, which was shown to increase abscission failure and accelerate hepatocytes hyperpolyploidization[13]. Physiological hepatocyte polyploidization has been proposed as a diversity factor for liver homeostasis, and as a mechanism to restrict proliferation and promote tumor suppression[14–16]. The consequences of pathological hyperpolyploidy, for example, when induced by chronic stress or exposure to carcinogens, are still largely unknown.

Under chronic stress, the adult liver has a remarkable ability to generate hyperpolyploid hepatocytes[17–20]. Recent study revealed that hyperpolyploid hepatocytes are associated with worse prognosis in human liver HCC, and that HCCs characterized by a low degree of differentiation and *TP53* mutations have higher levels of polyploidy[21]. The occurrence of hyperpolyploidy at early stages of tumor formation is believed to increase genomic instability and to be a pivotal step during tumorigenesis[22–24]. Hyperpolyploid giant cells, such as human ovarian, breast, colon, and prostate cancer cell lines, have been demonstrated to serve as a source of stemness and tumor heterogeneity through genomic reduction pathways[25]. Whether this is a shared feature of hyperpolyploid cells in different tissues is unknown.

Here, the pathological significance of genotoxin-induced hyperpolyploid hepatocytes around the centrilobular (CL) region is demonstrated. The CL hepatocytes, characterized by their ability to become hyperpolyploid, are the source of pre-neoplastic cells, likely through uncharacterized genome reduction processes. Upregulation of *Aurkb* is identified to cause abscission failure and promote hyperpolyploidization of hepatocytes. Our findings show that, under treatment with diethylnitrosamine (DEN), a carcinogenic compound known to cause HCC, hepatocytes hyperpolyploidization is a crucial step for the transformation of hepatocytes into HCC cells.

## Results

### DEN causes pathological hyperpolyploidization of CL hepatocytes.
To address how hepatocytes transform into HCC cells,

HCC was induced with DEN, and liver tissues were examined histologically at the preneoplastic stage. DEN-injected mice were sacrificed at the indicated times after 5 h BrdU-labeling (Fig. 1a). Global liver morphology showed no significant change up to 3 months after DEN injection, although very few small tumor nodules were sometimes found on the surface of DEN-treated liver (Fig. 1b). The hepatocytes from older control mice at p105 had larger nuclei compared to younger control mice, but DEN-treated hepatocytes still display significantly bigger cells and nuclei than age-matched control mice (Fig. 1c, d and Supplementary Fig. 1b–g), in line with previous findings[17]. Flow cytometry analysis also demonstrated that enlarged nuclei correlated with higher DNA content in DEN-treated hepatocytes (Supplementary Fig. 1d). Because the zonal distribution of hepatocytes has been proposed to be relevant to the pharmacokinetic of DEN[26], we next investigated whether the DEN-induced hyperpolyploid hepatocytes were zone-specific. Using glutamine synthetase (GS) expression as marker for CL hepatocytes[21], the central vein to portal vein (CV-PV) axis was subdivided into 15 parts (Supplementary Fig. 1a), and the sizes of cells and nuclei were quantified along this axis, as described previously[13]. Remarkably, only hepatocytes close to the CL and midlobular (ML) regions displayed enlarged nuclei compared to control liver (Fig. 1c, d and Supplementary Fig. 1c–g). Further examination of the effects of DEN on hepatocytes revealed these enlarged nuclei and cells became gradually evident around 2–3 months in DEN-treated liver (Fig. 1c, d and Supplementary Fig. 1f, g). In contrast, perilobular (PL) hepatocytes displayed similar nucleus sizes compared to age-matched control liver (Fig. 1c, d). To investigate whether hyperployploidization of CL hepatocytes is simply an artifact of the DEN model or a relevant feature of early-stage HCC formation caused by xenobiotics or oxidative stress, three different mouse models, using Aflatoxin B1 (AFTB1), carbon tertrachloride (CCl4), and 45 kcal% high-fat diet (HFD) were utilized to address this question. As shown in Supplementary Fig. 1k, hepatocytes in AFTB1- and CCl4-treated mice displayed enlarged nucleus size near the central vein region, compared to control group. After 90 days of HFD treatment, hepatocytes also exhibited slightly larger nucleus size than age-matched control nearby central vein region, but not in the liver with 60 days of HFD treatment. This result indicated that not only DEN, but also xenobiotics and HFD-induced oxidative stress specifically target CL and ML hepatocytes and cause hepatic hyperpolyploidization within these two regions. More generally, the metabolic sensitivity of CL and ML hepatocytes to xenobiotics is likely an important factor in HCC development.

To understand the cause of hyperpolyploidy, we next investigated whether cell proliferation or cytokinesis was affected by DEN. We recorded the number and distribution of BrdU-positive hepatocytes along the CV-PV axis. In control liver, the number of BrdU-positive hepatocytes progressively decreased during development, consistent with our previous finding (Fig. 1e, f and Supplementary Fig. 1h, i)[13]. In DEN-injected liver, however, the number of BrdU-positive hepatocytes increased (Fig. 1e, f). Importantly, the distribution of BrdU-positive hepatocytes was enriched around the CV, thus highly correlated with the distribution of hyperpolyploid hepatocytes (Fig. 1e, g and Supplementary Fig. 1h, i). As anticipated, BrdU-positive hepatocytes displayed enlarged nuclei, compared to age-matched control BrdU-positive hepatocytes (Supplementary Fig. 1j).

Next, primary hepatocytes isolated from liver tissues with or without DEN treatment were cultivated to follow the progression of cytokinesis by time-lapse microscopy (Fig. 1h). Mitosis successfully proceeded with no apparent abnormality, and the respective proportions of cytokinesis failure without contractile ring formation showed no differences between

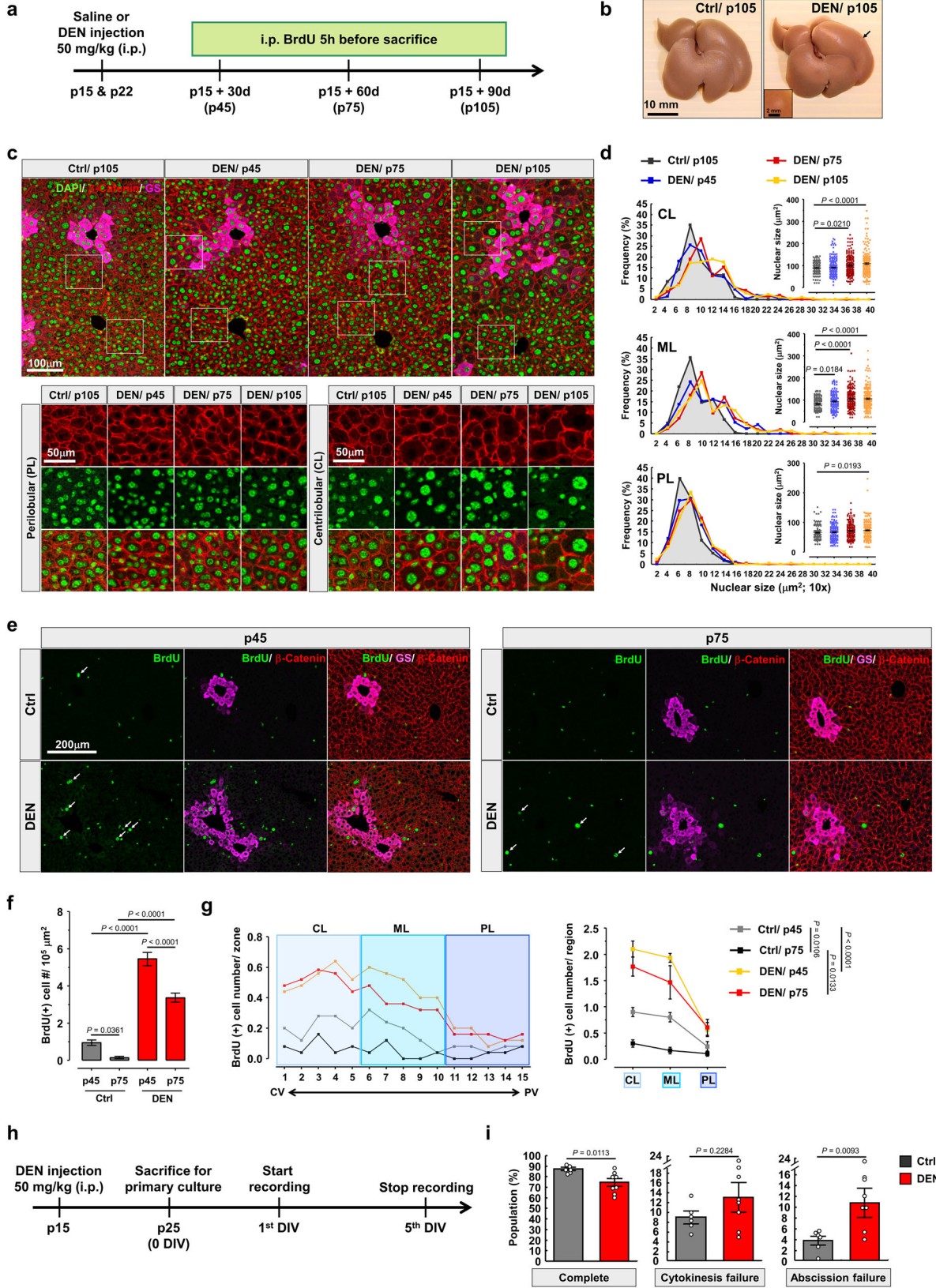

the two groups (Fig. 1i and Supplementary Movie 1, 2). However, after DEN treatment the frequency of abscission failure was significantly increased, in parallel with a reduced ratio of complete cytokinesis (Fig. 1i and Supplementary Movie 3). Interestingly, we found that bi-nucleated hepatocytes with enlarged nuclei were generated by two successive cycles with

abscission failure (Supplementary Movie 4), while hepatocytes with a single macronucleus were produced via abscission failure followed by complete cytokinesis (Supplementary Movie 5). Overall, these results indicated that aberrant cell proliferation coupled with abscission failure underlies DEN-induced hepatocyte hyperpolyploidization.

**Fig. 1 Emergence of hyperpolyploid hepatocytes within CL and ML regions of DEN-treated liver. a** Schema for DEN-injection experiment: Injection of DEN (50 mg/kg) or normal saline (Ctrl) was performed at p15 and p22. Mice were injected with BrdU, 5 h before sacrifice. **b** Example of images shows the global morphology of liver. **c** Representative images of liver sections from indicated mice. DEN-injected mice were sacrificed at indicated times, and control mice have the same age like mice with 3 months of DEN treatment. The higher magnification shows the detailed morphology of CL and PL hepatocytes. **d** The nucleus size of hepatocytes along CV-PV axis of the liver. Mouse number ($N$) = 5 mice/group from three independent experiments. Cell number ($n$) = 600, 630, 630 cells for each group, respectively. **e** Images of sections were prepared from p45 and p75 liver tissues with or without DEN injection, describing in (**a**). **f** The numbers of BrdU-positive hepatocytes in indicated liver at p45 and p75 age. Thirty different regions were quantified from $N$ = 5 mice/group. **g** Zonal distribution of BrdU-positive hepatocyte along CV-PV axis. CV-PV axis is divided into 15 parts. Part 1 to 5, 6 to 10, and 11 to 15 stand for CL, ML, and PL of the liver, respectively. $N$ = 5 mice/group. **h** Schematic of hepatocyte primary culture for time-lapse recording: Mice are injected with DEN or normal saline at p15 and sacrificed at p25 for primary culture. Twenty-four hours after seeding, the first day in vitro (DIV), hepatocytes were conducted to cell recording. **i** The ratio of indicated cytokinesis behaviors. Data are representative of at least three independent experiments. $n$ = 276 and 260 dividing cells for control and DEN-treated groups, respectively. Statistic: One-way ANOVA (**d**) and (**f**); two-way ANOVA (**g**); two-tailed Mann–Whitney nonparametric test (**i**). Values represent the mean ± SEM (also see Supplementary Fig. 1).

**Preoplastic lesions mainly originate within the CL region**. To investigate the correlation between hyperpolyploidy and the formation of preoplastic lesions, we first asked whether hyperpolyploid hepatocytes express precancerous markers. Nuclear vacuolation, a senescence marker identified as a precancerous phenotype[27–30], increased in DEN-injected livers (Fig. 2a and Supplementary Fig. 2a, b). Further analysis indicated that nuclear size was positively correlated with vacuolation number and area (Fig. 2b and Supplementary Fig. 2c). Importantly, nuclear vacuolation progressively increased after DEN injection (Fig. 2c), and hepatocytes with nuclear vacuolation were also highly enriched within the CL and ML regions, similar to the distribution of hyperpolyploid hepatocytes (Fig. 2d and Supplementary Fig. 2a, b). Furthermore, senescence and DNA damage were examined by Lamin B1 and γH2AX, respectively (Supplementary Fig. 2d–f). In DEN-treated liver, higher expression of γH2AX was detected specifically around CL hepatocytes and preoplastic foci, indicating severe DNA damage in these regions (Supplementary Fig. 2d, f). In contrast with γH2AX, downregulation of Lamin B1 was observed in DEN-treated liver, which was specifically identified in hepatocytes but not in other cell types such as portal triad endothelial cells (yellow arrow) (Supplementary Fig. 2e, f). Because hyperpolyploid hepatocytes with precancerous traits were mainly located within the CL and ML regions, we next sought to determine whether preoplastic lesions, a small cell change of the liver parenchyma with microscopic foci of altered hepatocytes (see definition in Supplementary material)[31,32], also predominantly occurred in these regions. We found that microscopic foci of preoplastic lesions were first observed 2 months after DEN injection, the number of preoplastic lesions increasing thereafter (Fig. 2e). These preoplastic lesions were not only BrdU- and Ki67-positive, but also expressed higher levels of α-fetoprotein, a precancerous marker of HCC, with higher nuclear-to-cytoplasmic ratio in area compared to normal cells (Fig. 2f–h, and Supplementary Fig. 2g). Importantly, the distribution of preoplastic lesions was indeed enriched around the CV, thus highly correlated with the distribution of hyperpolyploid hepatocytes (Fig. 2f, g, i and Supplementary Fig. 2g, h).

**Preoplastic cells are derived from CL hepatocytes and show nucleus size reduction**. Surprisingly, the preoplastic cells analyzed above were extremely small, and displayed smaller nuclei compared to cells in other areas of the liver, suggesting lower genomic content (Fig. 3a). Because reduced genomic content in human and rodent liver cancer cells has been previously demonstrated[33–35], we hypothesized that preoplastic cells originate from hyperpolyploid hepatocytes through genomic content and nucleus size reduction. If this is the case, preoplastic cells would appear predominantly among CL hepatocytes. To test this conjecture, CL and PL hepatocytes were

separately enriched by digitonin-collagenase perfusion system after 1 month of DEN treatment (Fig. 3b)[36,37], when hyperpolyploid hepatocytes emerge but before preoplastic foci formation. Digitonin caused a regularly scattered discoloration pattern on the liver surface and H&E-stained sections (Fig. 3c, d). In addition, expression of liver zone-specific markers confirmed the efficient enrichment of PL or CL hepatocytes (Fig. 3e). Importantly, microscopy and flow cytometry showed that CL hepatocytes did have bigger nuclei and higher DNA content compared to those isolated from the PL region after DEN treatment (Fig. 3f and Supplementary Fig. 3a, b).

To examine whether CL hepatocytes displayed cancerous properties after DEN injection, colony formation assay was performed with CL or PL hepatocytes, the numbers of colonies analyzed on the 21st day in vitro (DIV). CL hepatocytes showed higher colony formation ability compared to PL hepatocytes and to control hepatocytes isolated from the liver without DEN treatment (Fig. 3g, Supplementary Fig. 3c, and Supplementary Movie 6, 7). Further analysis of the nuclei demonstrated that colony-forming cells had significantly smaller nuclei compare to non-colony cells (Fig. 3h, i). Next, tumorsphere formation assay was applied to investigate whether CV hepatocytes were more like cancer stem/progenitor cells than PL hepatocytes after DEN injection. Most hepatocytes died within 48 h of seeding, but a population of spherical formations was observed 7 days after seeding. Huge tumorspheres from CL hepatocytes were easily distinguishable from aggregated cells on the 14th DIV and grew further until the 21th DIV (Supplementary Fig. 3d). Moreover, CL hepatocytes more frequently formed tumorspheres (diameter >100 μm) than PL hepatocytes (Supplementary Fig. 3e), and the tumorspheres derived from CL hepatocytes were larger than those from PL hepatocytes (Supplementary Fig. 3f). To address the tumorous potency of CL and PL hepatocytes after DEN treatment in vivo, subcutaneous implantation of spheres derived from CL- and PL-enriched hepatocytes into the right and left flanks of NSG™ immunodeficient mice was performed, respectively. Clumps were found in the right flanks (CL-enriched hepatocytes) 30 days post-injection, became significantly larger than those observed in the left flanks at 50 days of injection, and further grew thereafter, while no clumps were detected in the left flanks (Fig. 3j). H&E staining and immunohistochemistry revealed the severe angiogenesis and high expression of the cell proliferation marker Ki67 within the clumps, confirming the tumorigenicity of CL hepatocyte-derived spheres after DEN treatment (Fig. 3k and Supplementary Fig. 3g). To investigate whether hyperpolyploid hepatocytes are the origin of the smaller preoplastic cells, the size of the nuclei of cells in the clump was further analyzed. During isolation of CL hepatocytes, very few cells with small nuclei were observed, but 80 days after implantation cells in the CL hepatocytes-derived clumps had significantly smaller nuclei

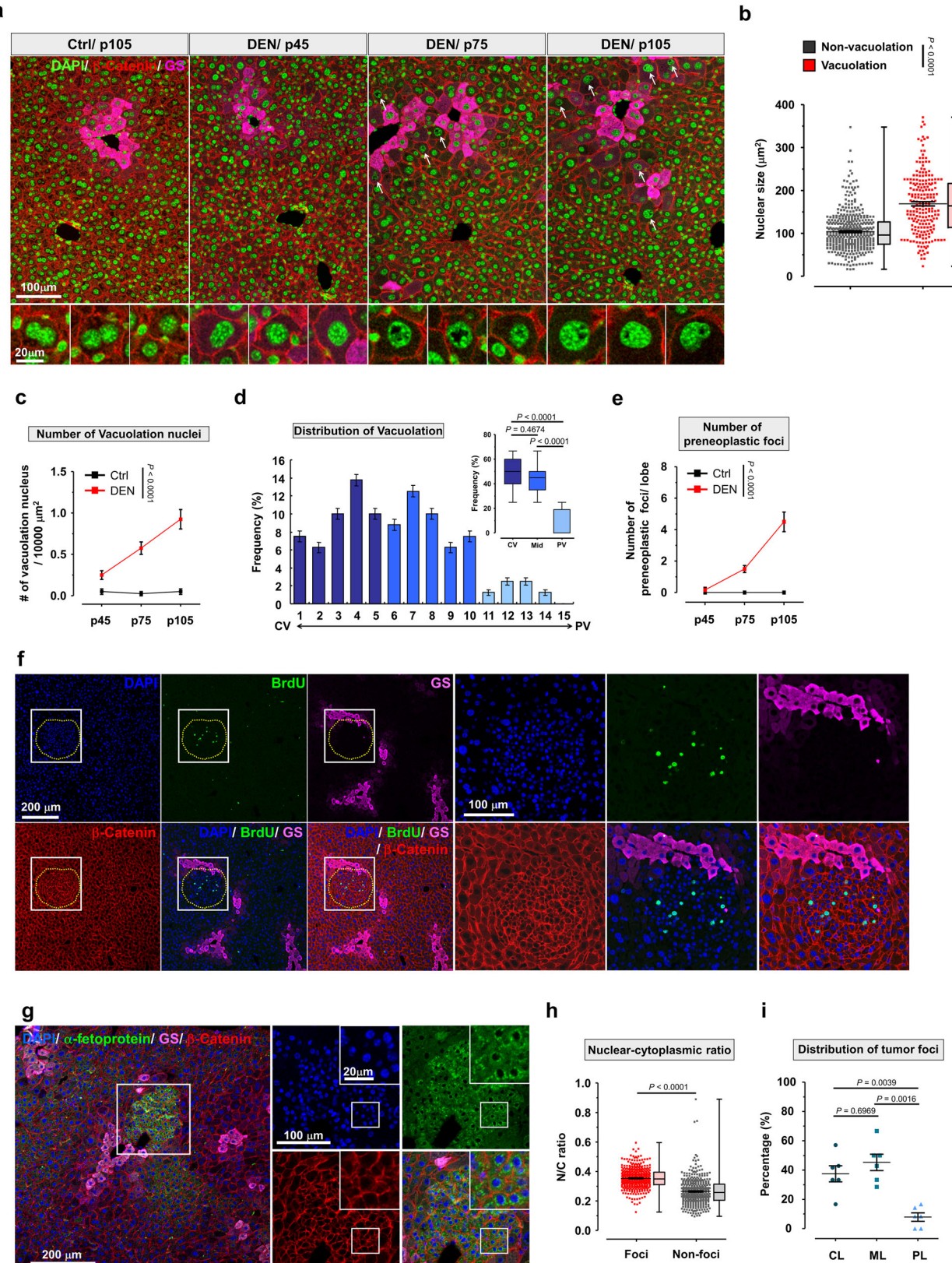

compared to CL hepatocytes in DEN-treated liver (Fig. 3l). Interestingly, multipolar dividing cells reported by Grompe's group as the reversion mechanism of polyploid hepatocytes were detected (around 5% incidence rate) within CL hepatocytes-derived tumors (higher magnification images of Fig. 3k)[38,39]. This observation raises the possibility that "reductive mitoses" can happen in CL hepatocyte-derived tumors and results in polyploidy reversal to produce daughter cells with lower DNA content. Together these results show that hyperpolyploid-enriched CL hepatocytes may have high tumorigenicity and are likely candidates for the origin of the small preneoplastic cells seen after DEN treatment.

**Fig. 2 CL and ML region are the dominant areas for preneoplastic lesions generation. a** Representative Images of liver sections from control (normal saline) and DEN-treated mice at indicated times. The high magnification shows the detailed morphology of nuclear vacuolation in hepatocytes. **b** The quantitative data of nucleus size of indicated hepatocytes. $N = 5$ mice/group, and $n = 420$ and $208$ nuclei from non-vacuolation and vacuolation hepatocytes, respectively. **c** The numbers of vacuolation nuclei in control and DEN-treated liver at indicated times. $N = 5$ mice/group. **d** Bar graph displays the frequency distribution of hepatocytes with nuclear vacuolation along the CV-PV axis in DEN-treated liver. Inset shows the statistic analysis of nuclear vacuolation between CL, ML, and PL region, $N = 5$ mice. **e** The numbers of preneoplastic foci in control and DEN-treated liver at indicated times. $N = 5$ mice/group. **f** Example of images shows the distribution of preneoplastic foci nearby GS positive hepatocytes. The high magnification represents the enrichment of BrdU-positive hepatocytes within preneoplastic foci. $N = 5$ mice in this treatment. **g** Images display the α-fetoprotein positive signal within preneoplastic foci. The high magnification images display the detailed information of fluoresce signals within preneoplastic foci. $N = 5$ mice in this examination. **h** The nuclear-cytoplasmic ratio of hepatocytes within the indicated region. $n > 400$ hepatocytes, 6 mice/group. **i** The emergence frequency of preneoplastic foci within the indicated liver lobule. $N = 6$ mice. Statistic: In (**b**), (**d**), and (**h**), data are represented as box plots where the middle line is the median, the lower and upper hinges correspond to the 25th and 75th quartiles, the upper whisker extends from the hinge to the largest value no further than 1.5 × IQR (interquartile range) from the hinge and the lower whisker extends from the hinge to the smallest value at most 1.5 × IQR of the hinge. Two-tailed Student's unpaired $t$-test (**b**) and (**h**); two-way ANOVA (**c**) and (**e**); one-way ANOVA (**d**) and (**i**). Values represent the mean ± SEM (also see Supplementary Fig. 2).

**Upregulation of *Aurkb* in DEN-injected liver, NAFLD, and HCC patients**. Because abscission failure is involved in DEN-induced hepatocyte hyperpolyploidization, we next sought to identify the mechanisms underlying abscission failure. Recently, high-throughput screens have identified critical genes involved in cytokinesis[40–42]. Of these 251 cytokinesis-related genes, 185 genes were also reported in two omics studies of DEN-regulated genes in rat liver (GSE19057 and GSE63726) (Fig. 4a and Supplementary Fig. 4a)[43,44]. Only 5 out of 185, including *Anxa2*, *Aurkb*, *Cdk1*, *Rhoc*, and *S100a6*, displayed a significant upregulation (>2-fold) in both datasets (Fig. 4b and Supplementary Fig. 4b). We confirmed by qPCR that four of these candidates exhibited a significant increase in mRNA expression in the liver after 1 month of DEN injections (Fig. 4c), with *Aurkb* and *Cdk1* increasing over five-fold. Interestingly, *Aurkb* is a master controller of the chromosomal passenger complex, and plays a key role in regulating the NoCut pathway of abscission[45–47]. Upregulation of AURKB is highly correlated with abscission failure, polyploidization, and tumor formation[45,46,48]. Focusing on this interesting candidate, we found that the expression and activity (by phosphorylation of Thr232, pT232) of AURKB increased in DEN-injected liver, together with Histone-H3 Ser10 phosphorylation, a direct target of AURKB (Fig. 4d).

To evaluate whether higher expression of *Aurkb* is positively correlated with human liver diseases, we first mined GEO datasets from human samples. Patients with simple steatosis and NAFLD in the GSE89632 dataset show a significant increase of *Aurkb* expression in the liver (Fig. 4e), echoing the similar findings of pathological polyploidization in NAFLD livers[20]. Furthermore, the expression of *Aurkb* gradually increased during liver fibrosis progression in GSE66232 dataset (Fig. 4e). In support of its putative oncogenic activity, significant upregulation of *Aurkb* was found in HCC patients from at least three GEO datasets (GSE20140, GSE54236, and GSE64041) (Fig. 4f), and *Aurkb* induction was confirmed in matching pairs of HCC and normal tissues in GES64041-3 dataset (Supplementary Fig. 4c). Importantly, *Aurkb* expression showed significantly negative correlation with tumor doubling time and patient survival (Fig. 4g, h and Supplementary Fig. 4d). These results efficiently correlate higher expression of *Aurkb* with the progression of human liver diseases.

**Upregulation of pT232-AURKB causes abscission failure in DEN-treated hepatocytes**. Since activity and subcellular distribution play critical roles in modulating the functions of AURKB[47], we next investigated whether DEN treatment affected the subcellular localization of AURKB. In hepatocyte primary cultures (Fig. 5a), pT232-AURKB showed typical subcellular distribution in control and DEN-treated hepatocytes during the

cell cycle (Supplementary Fig. 5a). Consistent with previous findings, we observed that pT232-AURKB relocated from the central spindle to the midbody sequentially, pT232-AURKB levels gradually decreasing to be almost undetectable during abscission in control hepatocytes[45,46]. In stark contrast, DEN-treated hepatocytes showed higher levels of pT232-AURKB at all mitotic stages, compared to control hepatocytes (Supplementary Fig. 5a). Both AURKB and pT232-AURKB displayed higher intensity and wider distribution at the midbody in DEN-treated hepatocytes compared to control during abscission (Fig. 5b, c and Supplementary Fig. 5b, c). In addition, longer intercellular bridges between daughter cells were observed in DEN-treated hepatocytes (Fig. 5b, d), resulting from the uncut intercellular bridge being pulled in different directions by the two daughter cells[13,49]. Regression analysis suggested that the length of intercellular bridge is positively correlated with the intensity of AURKB and pT232-AURKB at the midbody (Supplementary Fig. 5d). To test the causal relationship between the pT232-AURKB level at the midbody and intercellular bridges length, phosphorylation of AURKB was inhibited with AZD1152 (Barasertib), a highly specific AURKB inhibitor, in DEN-treated hepatocytes. A dose-dependent decrease of pT232-AURKB was confirmed in cultured hepatocytes under AZD1152 treatment (Supplementary Fig. 5e). Signal normalization of pT232-AURKB by AURKB suggested that only pT232-AURKB, not total AURKB levels, decreased under AZD1152 treatment at the midbody (Fig. 5b, c and Supplementary Fig. 5b), accompanied as hypothesized by a significantly reduced length of intercellular bridges in DEN-treated hepatocytes (Fig. 5b, d). Critically, the proportion of cultured hepatocytes failing abscission after DEN treatment decreased significantly in the presence of AZD1152 (Fig. 5e). Altogether, these results show that DEN induces expression and activity of AURKB in mouse liver, facilitating abscission failure of hepatocytes.

**Reducing AURKB activity ameliorates DEN-induced hyper-polyploidy**. High levels of pT232-AURKB have been reported to restrain abscission, leading to polyploidy through stabilization of intercellular canals[45,46]. We thus addressed whether hyper-activation of AURKB is involved in mediating DEN-induced nuclear enlargement in vitro. We first measured the size of nuclei during 3 consecutive days of primary hepatocyte cultures in the presence of vehicle, DEN, or DEN + AXD1152 (Supplementary Fig. 6a), showing a progressive increase in all groups (Fig. 6a–c). The largest nuclei were observed in DEN-treated hepatocytes compared to control group, consistent with our previous data (Figs. 1c, d and 6a–c). Under DEN + AZD1152 treatment, nuclear size was reduced compared to DEN-treated hepatocytes,

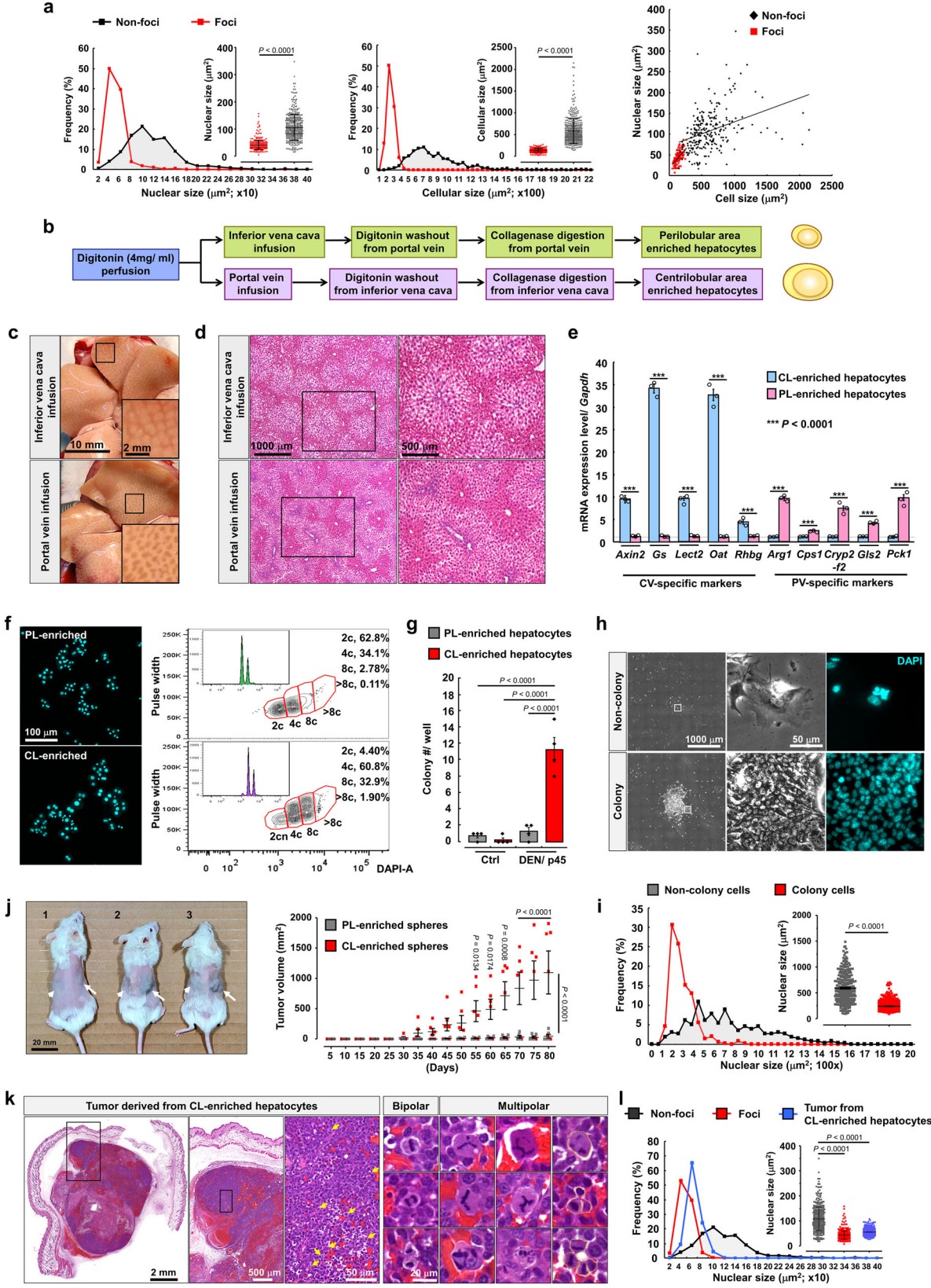

with AZD1152 showing a significant time-course effect (Fig. 6a–c). This result suggested that the level of pT232-AURKB plays a critical role in controlling hyperpolyploidization of hepatocytes.

Because genomic reducing events have been demonstrated as a predictive sign of stemness and genetic instability during tumorigenesis[25], we then performed a systemic survey of nuclear morphology. DEN-treated hepatocytes displayed higher micronuclei and nuclear budding ratio than in control group, AZD1152 treatment decreasing this ratio (Fig. 6d, e and Supplementary Fig. 6b). The number and size of micronuclei in DEN-treated hepatocytes were also increased, and significantly reduced after

**Fig. 3 Preneoplastic cells are derived from hyperpolyploidy-enriched CL region. a** Nucleus and cell size within preneoplastic foci and non-preneoplastic region. $n > 400$ hepatocytes/group, $N = 6$ mice/group. **b** Schema for isolation of PL- and CL-enriched hepatocytes by digitonin-collagenase perfusion system. **c** Representative images display discoloration pattern on liver surface after digitonin infusion from inferior vena cava or portal vein, respectively. **d** Representative images show the H&E staining of indicated liver histologically. Digitonin infusion caused discoloration of eosin signal. $N = 4$ mice/group. **e** qPCR analysis of mRNA levels of CL- and PL-specific markers. $N = 3$ mice/group. **f** The nucleus morphology of hepatocytes isolated by digitonin-collagenase infusion. The DNA contents of CL- and PL-enriched hepatocytes were analyzed by flow cytometry. $N = 4$ mice/group. **g** The colony numbers at 21th day in vitro (DIV) in indicated cultured hepatocytes. $n = 4$ independent experiments. **h** Representative images of colony and non-colony hepatocytes morphology isolated from CL region at 21th DIV. Low and high magnification show the morphology of colonies and nuclei, respectively. $n = 4$ independent experiments. **i** Nucleus size in colony and non-colony hepatocytes at 21th DIV. $n = 500$ and 257 cells from four independent experiments, respectively. **j** Visible differences in tumor size under the skin of the dorsal body surface in NSG™ mice after 10 weeks of subcutaneous transplantation. Arrow and arrow head indicate subcutaneous implantation of hepatic spheres derived from CL- and PL-enriched hepatocytes, respectively. Enlarged tumors were observed in CL-enriched hepatocytes transplanted region in j-2 and -3 mice. No tumors were detected in j-1 mouse. Line graphs show the tumor growth cure of subcutaneous transplants. $N = 6$ mice. **k** Example of images show H&E staining of tumor derived from CL-enriched hepatocytes under low and high magnification. Arrows (cyan) indicates the proliferating tumor cells at metaphase. The highest images display bipolar and multipolar dividing cells in tumor nodes. Dashed lines delineate cell borders. $N = 6$ mice for each implantation. **l** Line and dot graph show the quantitative data of nucleus size within preneoplastic foci, non-preneoplastic region and tumor derived from CL-enriched hepatocytes. $n = 420$, 408, and 420 cells, respectively. Statistic: Two-tailed Student's unpaired $t$-test (**a**, **i**); two-tailed Mann–Whitney nonparametric test (**e**); one-way ANOVA (**g**) and (**l**); two-way ANOVA (**j**). Values represent the mean ± SEM (also see Supplementary Fig. 3).

AZD1152 treatment (Fig. 6d, f and Supplementary Fig. 6c, d). Importantly, micronuclei and nuclear budding were also observed in CL hepatocytes with large nuclei in DEN-injected liver (Supplementary Fig. 6e), but were very rare in control liver, consistent with our in vitro results (Fig. 6d–f).

Together with our findings presented in Fig. 3, these results paint a picture in which aberrant AURKB-induced hyperpolyploidy also plays a role in the transformation of CL hepatocytes into preneoplastic cells via subsequent reduction of nucleus size, the sign of a probable genome reduction event. We further examined whether progeny cells with smaller nucleus and expressing cancer stem cell (CSC) markers could be detected in cultured hepatocyte after DEN treatment. Very few cells co-expressing the CSC markers EpCAM and Vimentin were observed in vitro but were significantly more abundant in hepatocytes cultured from DEN-treated liver compared to control, and decreased under AZD1152 treatment (Supplementary Fig. 6f, g). Altogether, our observations demonstrate that aberrant activation of AURKB is a critical pathway leading to DEN-induced hepatocyte hyperpolyploidization, promoting nuclear budding, and micronuclei progression.

**Pharmacological blockade of pT232-AURKB reduces preneoplastic lesion formation in DEN-injected liver.** We wondered whether halting hepatocyte hyperpolyploidization could be accompanied by attenuated preneoplastic lesions formation in DEN-treated liver. AZD1152 was injected intraperitoneally every 2 days starting 1 week after DEN injection for pathological examination (Supplementary Fig. 7a). DEN injection dramatically increased AURKB, but subsequent AZD1152 treatment reduced pT232-AURKB and its downstream target to control levels (Supplementary Fig. 7b). We next addressed the effect of AZD1152 on the number of BrdU-positive cells. The numbers of BrdU-positive hepatocyte and non-hepatocytes were analyzed, revealing AZD1152 significantly reduced the number of BrdU-positive hepatocytes in a dose-dependent manner, compared to DEN-treated liver. However, no significant effects of AZD1152 were seen on the proliferation of non-hepatocytes (Fig. 7a, b). Next, we examined the effect of AZD1152 on nucleus and cell size along the CV-PV axis. Like our previous findings (Fig. 1c, d and Supplementary Fig. 1c–e), DEN treatment caused hyperpolyploidization of CL and ML hepatocytes predominantly. Injections of AZD1152 and DEN significantly reduced the level of hyperpolyploidization (Fig. 7c–e and Supplementary Fig. 7c, d). We

further investigated whether AZD1152 rescued the extent of nuclear vacuolation in DEN-treated liver. A higher frequency of nuclear vacuolation was detected DEN-treated liver; after AZD1152 treatment, however, the number and size of nuclear vacuolation were significantly reduced, which indicated the suppressive effect of AZD1152 on the DEN-induced appearance of the precancerous phenotype (Fig. 7f). To evaluate the suppressive effect of AZD1152 on preneoplastic lesions formation, we investigated the pathological morphology of liver tissues globally and microscopically. After 3 months of DEN injections, no obvious changes in the general morphology of the liver were seen compared to control liver. Small tumor nodules were sometimes observed on the surface of DEN-treated livers (Supplementary Fig. 7e), but not with AZD1152 (Supplementary Fig. 7e). We then examined H&E-stained liver sections microscopically, revealing that AZD1152 treatment significantly decreased the number of preneoplastic foci in DEN-treated liver (Fig. 7g). Moreover, AZD1152 treatment also partially reduced the size of the larger tumor nodules observed on the surface of the liver 6 months after DEN treatment (Fig. 7h, i).

Altogether, these data show that the DEN-induced emergence of preneoplastic hepatocytes at least partially depends on AURKB-mediated hyperpolyploidization. Critically, the transformation of hepatocytes into HCC cells can be prevented by AZD1152, once more underlining the potential of this inhibitor as a chemotherapeutic drug.

**Discussion**

Hepatocarcinogenesis is a multistep progression with genetic alterations that lead to malignant transformation of hepatocytes[3], but how hepatocytes transformation occurs remains unresolved. Understanding the transformation process of hepatocytes into tumor cells is a critical issue to improve early clinical diagnosis of HCC and prevent hepatocarcinogenesis at an early stage. A significant feature observed in human and rodent liver tumor is a dramatic modification of the ploidy distribution of hepatocytes within the tumor nodules and in the noncancerous portions of the liver[21,33–35]; however, these data were obtained at the cancerous stage, not in precancerous livers, offering little insight on the processes occurring during transformation. Here, we used a well-described HCC mouse model but focused on the initiation stage, demonstrating the mechanisms underlying hepatocyte transformation systemically (Fig. 8). Microscopic foci of preneoplastic lesions, based on previous definition[31,32], adjacent the

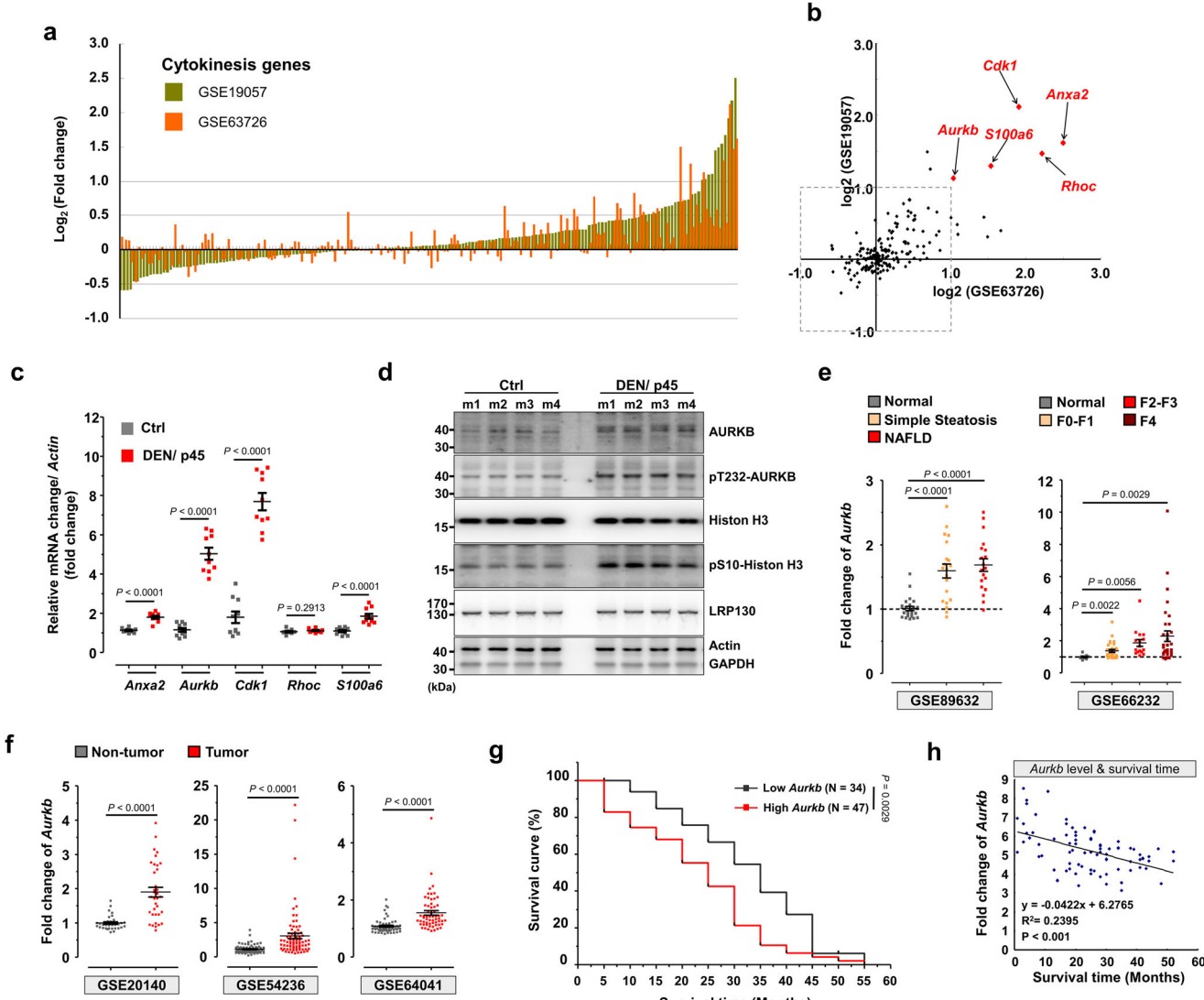

**Fig. 4 Upregulation of *Aurkb* in patients with liver diseases. a** The rank order of mRNA expression levels of cytokinesis genes from GEO datasets (GSE19057 and GSE63726). **b** Scatter plot graph with logarithmic axes shows the top five cytokinesis genes upregulated simultaneously in two GEO datasets. **c** qPCR analysis of mRNA expression levels of the top five cytokinesis genes in the liver. $N = 10$ mice/group. **d** Immunoblot detection of AURKB and pT232-AURKB levels in indicated mouse liver tissues. $N = 4$ mice/group. **e** The expression levels of *Aurkb* from human liver biopsies in GEO datasets. Steatosis and NAFLD patients (left panel), different fibrosis stage of human livers (right panel). Patient numbers: GES89632, $N = 24/20/19$; GSE66232, $N = 10/32/16/33$. **f** Human *Aurkb* expression in tumor and non-tumor biopsies from GEO datasets. Patient numbers: GES20140, $N = 34/35$; GSE54236, $N = 80/81$; GSE64041, $N = 60/60$. **g** Kaplan–Meier plots for survival curves of patients with low and high *Aurkb* expression in the liver. Patient numbers: GES54236, $N = 44/37$. **h** The linear regression analysis of the correlation between *Aurkb* expression and survival time of patients. Each point represents an individual case from a patient. Patient numbers: GES54236, $N = 81$. Statistic: Mann–Whitney nonparametric test (**c**); one-way ANOVA (**e**); two-tailed Student's unpaired *t*-test (**f**); log-rank (Mantel–Cox test) (**g**); linear regression (**h**). Values represent the mean ± SEM (also see Supplementary Fig. 4).

CV marker were observed in DEN-injected liver, surrounded by hyperpolyploid hepatocytes (Figs. 1 and 2), in line with previous studies[28]. Deficiencies in circadian genes also increased abscission failure-mediated CL hepatocytes hyperpolyploidy and liver tumorigenesis[13,50], suggesting a possible link between the formation of hyperpolyploid hepatocytes and an early stage of HCC progression. Moreover, mouse models have established a relationship between stresses and pathological hyperpolyploidy in the liver[17–20], but direct evidence of tumorigenesis are lacking. Our study demonstrated that xenobiotics (DEN, AFTB1, and CCl₄) and HFD can similarly induce hyperpolyploidization of CL hepatocytes, which may be caused by pericentral injury[26,51,52]. These models suggested that insults to CL hepatocytes is a common pathway leading to hyperpolyploidization, but the

mechanisms underlying early carcinogenesis in these different models may nevertheless be different and remain to be investigated. Most importantly, our results reveal a molecular correlation between pathological hyperpolyploidy of CL hepatocytes and DEN-induced HCC formation in mouse.

The expression and activity of AURKB are tightly controlled, either up- or downregulation of AURKB causes chromosomal instability, abscission failure, and polyploidization[47]. Upregulation of *Aurkb* is highly correlated with tumorigenesis of HCC[48,53,54]. Here, we have demonstrated that upregulation of AURKB plays a critical role in promoting abscission failure and hepatocyte hyperpolyploidization. Because endoreplication of hepatocytes has been also described under pathological conditions such as non-alcoholic fatty liver disease or liver cancer[12], we

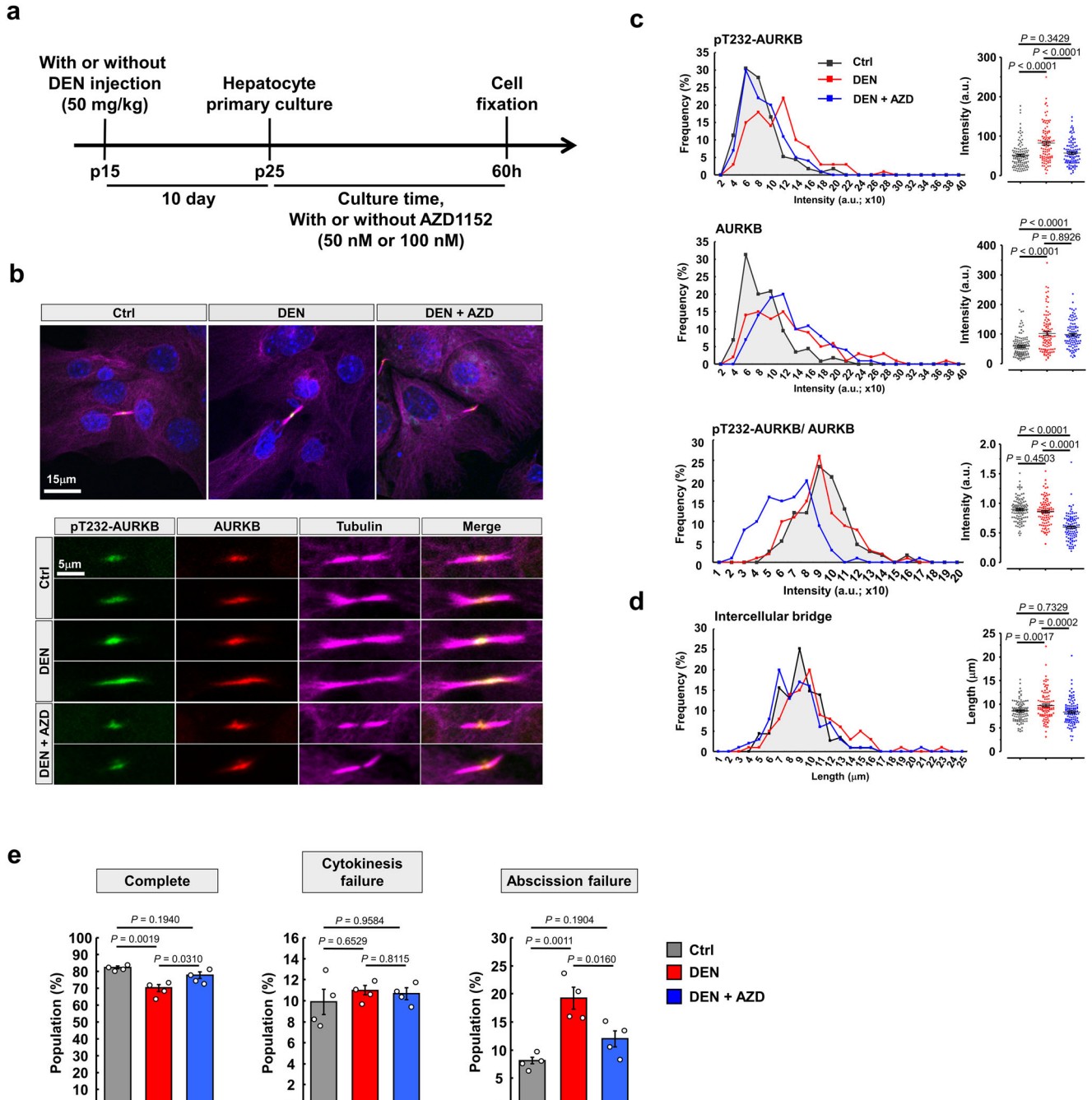

**Fig. 5 Hyperactivation of AURKB mediates DEN-induced abscission failure. a** Schema for detection of AURKB and pT232-AURKB in hepatocyte primary culture: Injection of normal saline or DEN at p15 followed by mice sacrifice at p25 for primary culture. Cultured hepatocytes were treated with AZD1152 or equal amount DMSO. **b** Immunofluorescence of pT232-AURKB and AURKB during abscission in indicated cultured hepatocytes with or without 50 nM AZD1152 treatment. The morphology of nuclei and intercellular bridges were outline by counterstaining with tubulin and DAPI, respectively. High magnification shows the detailed structure of the midbodies. **c, d** Quantification of the intensity of pT232-AURKB, AURKB, and AURKB normalized pT232-AURKB at the midbody from (**b**). The length of intercellular bridge was quantified by analyzed the highly condensed tubulin signal between daughter cells. $n = 115$, 100, and 100 dividing hepatocytes at abscission stage, respectively, per group from three independent experiments. **e** Quantification of the abscission failure ratio in the indicated cultured hepatocytes from time-lapse images. $n > 500$ dividing hepatocytes/group, four independent experiments/group. Statistic: one-way ANOVA (**c–e**). Values represent the mean ± SEM (also see Supplementary Fig. 5).

cannot exclude the contribution of endoreplication in hepatocyte hyperpolyploidization.

According to in vitro and in vivo assays presented in Fig. 3g–l, our findings support the hypothesis that CL hepatocytes, after a reductive mitosis or other uncharacterized genome reduction events[25,38,39], may be the origin of HCC cells. In comparison with

the smaller size PL-enriched hepatocytes, CL-enriched hepatocytes indeed display higher potential to generate spheres and tumor cells, although it is not possible to exclude that some other low-abundance cells included in the CL fraction during isolation may also have the opportunity to give rise to tumor cells. A recent study indicated that "normal" polyploidy plays a

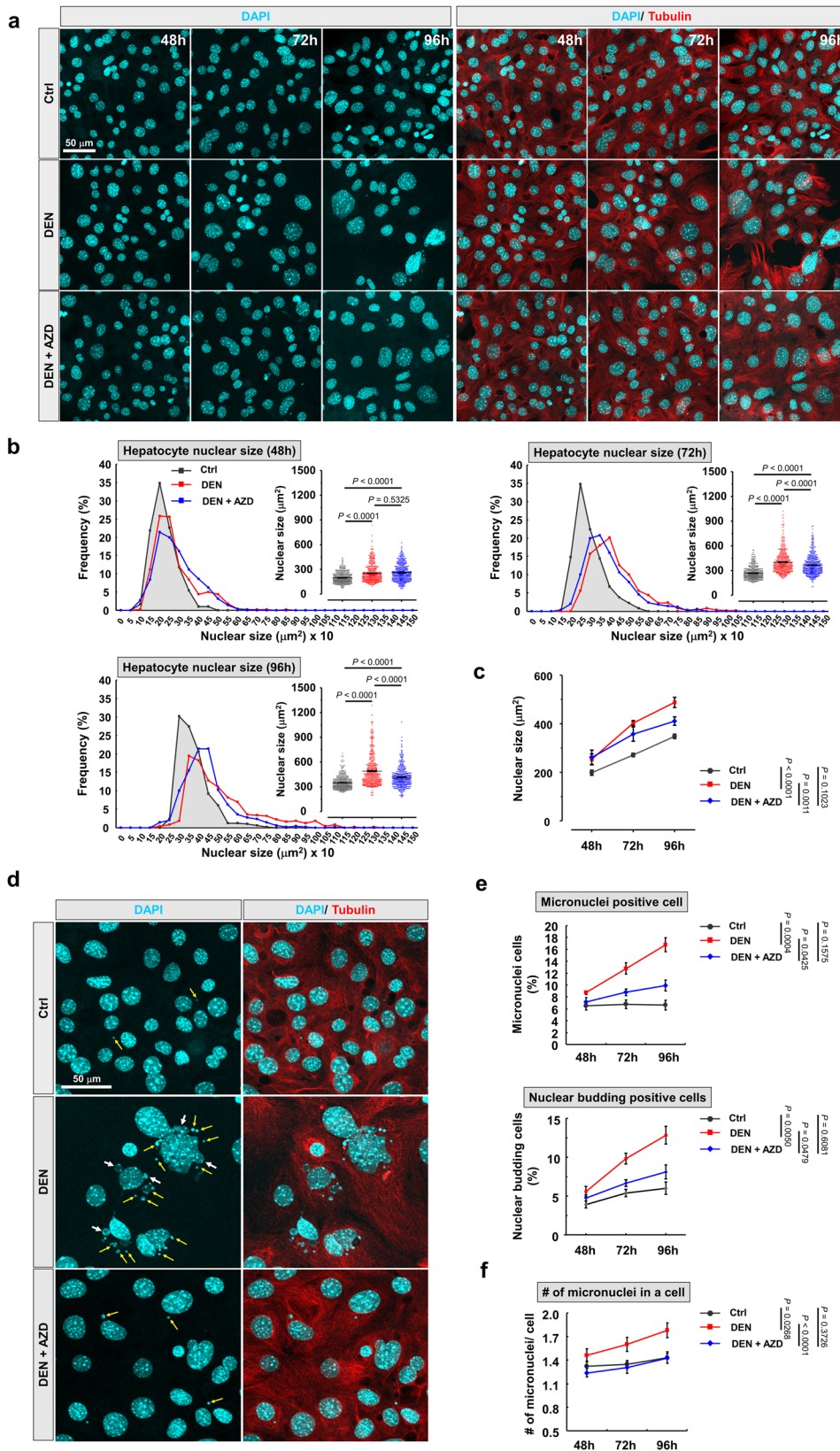

tumor-suppressive role in the liver, but pathological hyperpolyploidy under stress conditions is a different situation[16]. We hypothesize that, under normal conditions, polyploidy indeed suppresses tumorigenesis. Under stress, hyperpolyploid hepatocytes may initially represent an adaptive and protective response. Beyond a bearable (and maybe reversible) level of stress and as genomic content further increase, hyperpolyploid hepatocytes may lose the ability to preserve their genomic stability,

**Fig. 6 Hyperactivation of AURKB drives DEN-induced hyperpolyploidization of hepatocytes. a** Pharmacological manipulations of AURKB activity, according to the schema in Supplementary Fig. 6a, changes the nucleus size of cultured hepatocytes. Cultured hepatocytes were treated with 50 nM AZD1152 or equal amount DMSO. Nucleus and cell morphology were outlined by DAPI (cyan) and tubulin (red), respectively. **b, c** Frequency distribution and dot plot graphs display the nuclear sizes of indicated cultured hepatocytes at different time points. Line graph shows the nucleus size in indicated cultured hepatocyte between 48 and 96 h. $n = 500$ hepatocytes/group from four independent experiments. **d** Representative images show nuclear budding (white arrow) and micronuclei (yellow arrow) in indicated cultured hepatocytes. **e, f** The percentage of micronuclei and nuclear budding positive cell and the number of micronuclei in a cell of indicated cultured hepatocytes. $n = 100/150/150$ (Ctrl), 125/150/150 (DEN), and 150/200/200 (DEN + AZD) cells, respectively. Statistic: one-way ANOVA (**b**); two-way ANOVA (**c**), (**e**), and (**f**). Values represent the mean ± SEM (also see Supplementary Fig. 6).

ultimately leading to a higher frequency of genomic alterations and tumorigenesis[24]. Indeed, the nucleus size of CL-enriched hyperpolyploid hepatocytes decreased, preceding the formation of tumor colonies (Fig. 3), echoing the findings of near-diploid HCC cells in human and rodent tumor nodules[33–35]. The proposed role of polyploid giant cancer cells in metastasis[25] and the reversion of polyploid hepatocytes via the ploidy-conveyor[38,39] also support the hypothesis of a genome reduction event in hyperpolyploid hepatocytes in the formation of preneoplastic lesions. Our results do not exclude the possible contribution of different stem/progenitor populations that may respond to oncogenic or growth-promoting signals secreted from hyperpolyploid hepatocyte. Although we did not confirm whether a genome reduction actually occurred, we observed multipolar dividing in CL hepatocytes-derived tumor cells, which may be an indication of a genome reduction event (Fig. 3k). Moreover, frequent nuclear budding and micronuclei in hyperpolyploid hepatocytes were identified in vitro (Fig. 6), and under DEN treatment CSC markers-positive cells were often small and had small nuclei, a pathological phenotype that improved with AZD1152 (Supplementary Fig. 6e–g). Thus, based on our data, it is reasonable to speculate that hyperpolyploid hepatocytes, through reduction of genomic content, may play a source of stemness and tumor heterogeneity during initiation stage. The mechanisms underlying the transformation of hyperpolyploid hepatocytes into preneoplastic cells with low DNA content, and the impact of subgenomic content transmission remain to be investigated.

In conclusion, our data strongly suggest that after DEN-induced carcinogenic liver injury, hyperpolyploid hepatocytes arising from abscission failure are oncogenerative cells and a major source of preneoplastic lesions. These discoveries provide a significant contribution in understanding of liver cancer biology, and may lead to new avenues in identifying potential pathological markers for early diagnosis and prevention of hepatocarcinogenesis.

## Methods

**Antibodies and chemicals**. Antibodies used in detecting specific proteins are BrdU (GeneTex, GTX26326, clone# BU1/75, 500x for IHC), GAPDH (Santa Cruz, sc-25778, clone# FL-335, 1000x for WB), β-actin (Sigma-Aldrich, A5441, clone# AC-15, 1000x for WB), α-tubulin (Sigma-Aldrich, T5168, clone# B-5-1-2, 1000x for WB), β-Catenin (Santa Cruz, sc-7199, clone# H-100, 500x for IHC), Glutamine Synthetase (BD, 610518, lot# 4357628, 1000x for IHC), AURKB (Abcam, ab2254, lot# GR3210135-1, 1000x for WB), pT232-AURKB (Rockland, 660-401-667, lot# 30691, 1000x for WB), Histone H3 (ABclonal, 2914, lot# 9, 1000x for WB), Histone H3 (ABclonal, A2348, lot# 350237701, 1000x for WB), pS10-Histone H3 (Cell Signaling, 9701, lot# 17, 1000x for WB, 9713) α-fetoprotein (Santa Cruz, sc-130302, clone# 39, lot# B1518, 250x for IHC), EpCAM (Santa Cruz, sc-53532, clone# G8.8, lot# H167, 250x for IHC), Lamin B1 (ABclonal, A1910, lot# 3560261001, 1000x for IHC), and γH2AX (ABclonal, AP0687, lot# 4000000110, 1000x for IHC). For immunofluorescence, AlexaFluor 488, 594, or 647-conjugated secondary antibodies are from Invitrogen. For immunoblot assay, horseradish peroxidase (HRP)-conjugated secondary antibodies were from Invitrogen. Chemicals used in the study are Corning Matrigel Matrix (CORNING, 354234), ProLong antifade, and DAPI reagents from Invitrogen. AZD1152-HQPA (SML0268), Avertin (T48402), and diethylnitrosamine (55-18-5) were purchased from Sigma-Aldrich. BrdU (REF 000103/LOT 1923353A; 3 mg/ml) was from Life Technologies.

**Experimental animals and drugs treatment**. All mice used for experiments were standard male ICR strain exception of special cases. Female NSG™ immunodeficient mice (The Jackson Laboratory, stock No: 005557) with 8 weeks age were conducted for subcutaneous implantation. For high-fat diet treatment, wild-type C57BL/6J mice were utilized, which are susceptible to diet-induced obesity at 8 weeks age. Mice were maintained in 12 h light/12 h dark cycle (LD) with food and water ad libitum. Temperatures of 20–23 °C with 40–60% humidity are maintained. Before perfusion and sacrifice, mice were anesthetized with Avertin (0.25 mg/g body weight). For observation of DEN-induced preneoplastic foci, mice were double injected (i.p.) with DEN (50 mg/kg body weight) at 15 and 22 days age and sacrificed at indicated times. For Aflatoxin B1 (AFTB1) model, male mice were double injected with a 6 mg/kg body weight dose of AFTB1 dissolved in DMSO at 15 and 22 days age and sacrificed at indicated times. For CCl₄ chronic-injury model, male mice with 15 days age were injected with a 0.5 mg/kg body weight dose of CCl₄ dissolved in corn oil every 3 days for 10 doses to induce chronic damage. For high-fat diet model, control and experimental groups were treated with 8 kcal% normal chow diet and 45 kcal% high-fat diet at 8 weeks age for indicated times, respectively. For time-lapse recording of hepatocytes in vitro, single-dose injection of DEN (50 mg/kg body weight) or saline was applied through i.p. to 15-day-old mice, and mice were then sacrifice at 25 days age for hepatocyte primary culture. The time-lapse recording was initiated after 24 h of inoculation (see time-lapse recording section for detailed information). For DEN time-series experiment, 50 mg/kg DEN in saline was injected i.p. twice at p15 and p22, and saline was applied into control group. Mice were sacrificed after 1, 2, and 3 months after DEN injection for further examination. Five hours before sacrifice, BrdU (10 μl/g body weight; 3 mg/ml) purchased from company was directly used and injected i.p. for cell proliferation analysis. To suppress AURKB activity in vivo, AZD1152 (25 or 50 mg/kg body weight) was injected 1 week after DEN injection through i.p. every 2 days for 1 or 2 months. AZD1152 powder was dissolved in DMSO to get the stock with concentration of 10 mg/ml. Then, AZD1152 was finally diluted into working concentration by normal saline for i.p. injection. Equal amount DMSO in saline was applied to control and DEN groups. Mice were sacrificed after 1, 2, 3, and 6 months of DEN injection for further examination. For cell proliferation analysis in vivo, BrdU was injected i.p. for 5 h before mice sacrifice. All animal experiments were approved by the animal experimentation committee of Taipei Medical University and performed in accordance with the guidelines of the institutional committee for the use of animals for research. The different groups were housed together in the same cages in all animal experiments. Animal studies were approved by the animal experimentation committee of Taipei Medical University and performed in accordance with the guidelines of the institutional committee for the use of animals for research. IACUC at TMU Approval No: LAC-20l6-0323.

**Human sample analysis**. To make a link between biomedical discovery in mouse model and clinical findings, genomic alterations of Aurkb in various clinical cases were obtained from Gene Expression Omnibus (GEO) datasets in NCBI. Genomic expression profiles of human liver biopsies were extracted from Series Matrix Files of GEO datasets (http://www.ncbi.nlm.nih.gov/geo/). The fold change of Aurkb in patients with steatosis, NAFLD, and various stage of fibrosis was acquired from GSE89632 and GSE66232. Examination of Aurkb expression in tumor and non-tumor biopsies of human liver was obtained from GSE20140, GSE54236, and GSE64041. The correlation between Aurkb expression level and survival time of patients was analyzed according to GSE54236. The case numbers and statistical analysis were display in the figure legends. Accession codes with hyperlinks of human GEO datasets are listed in "Data availability" section.

**Hepatocyte primary culture**. Primary mouse hepatocytes were isolated from 3-week-old mouse livers using a two-step collagenase I perfusion protocol. Briefly, liver was perfused with Ca²⁺- and Mg²⁺-free Hank's balanced salt solution (HBSS) from inferior vena cava with a flow rate of 5 ml per min for 3 min at RT. An incision was made in the portal vein to let blood out of the liver. Liver was then perfused with 10 mg/ml collagenase I (Worthington Biochemical Corporation) in Ca²⁺- and Mg²⁺-positive HBSS (Gibco) with a flow rate of 3 ml per min for 10 min. After perfusion, liver was disrupted with scissors and dissociated by pipetting gently in DMEM medium. Cell suspension was filtered through a cell strainer (100 μm²) and mixed with an equal volume of 90% Percoll (Sigma) in

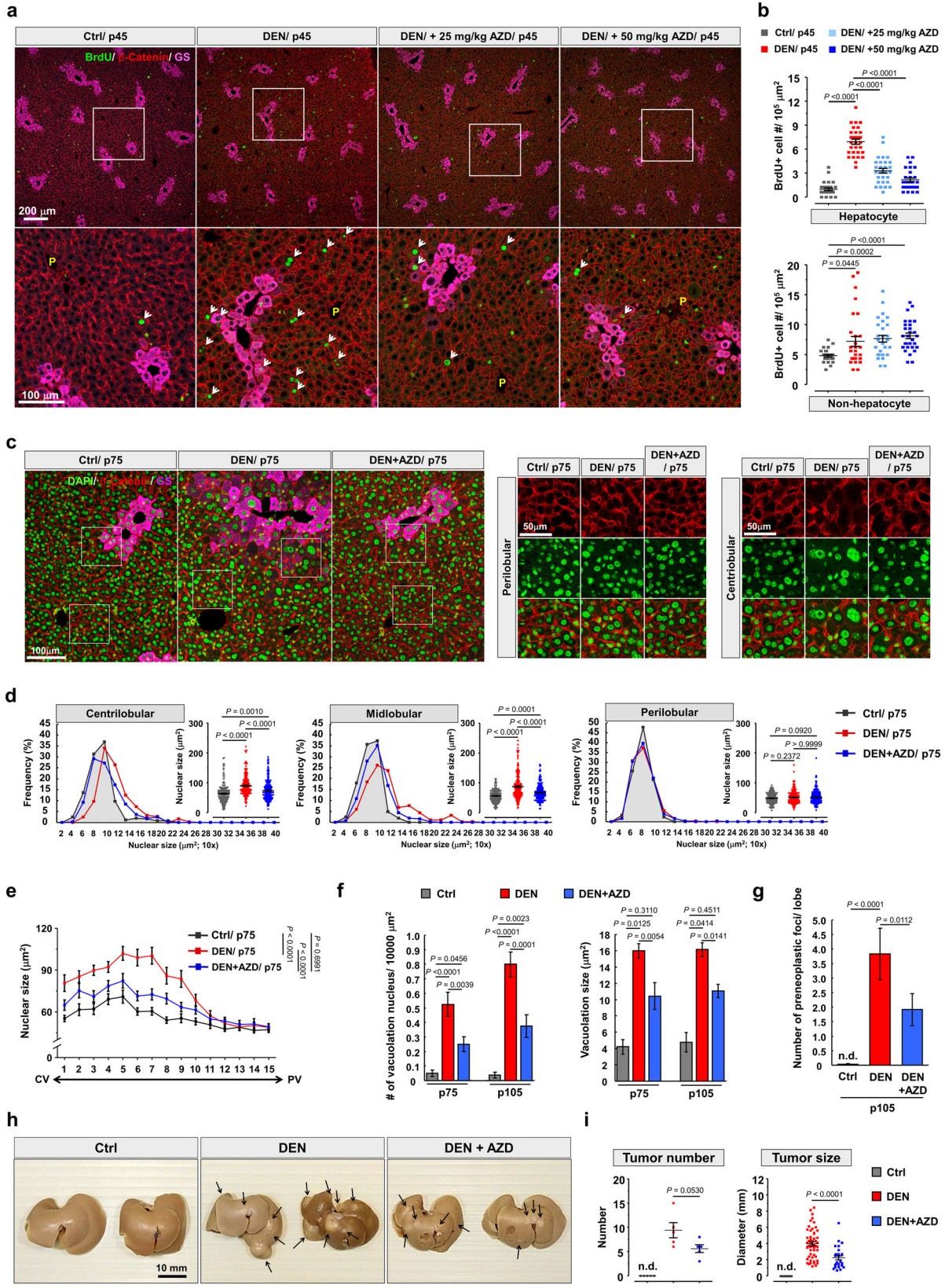

HBSS. To collect live hepatocytes, the mixture was centrifuged at $100 \times g$ for 5 min, and the pellet was resuspended in 199 medium (Gibco) containing 5% FBS. Isolated hepatocytes were seeded on collagen-coated 6-well plate (density = $2 \times 10^5$) or 12-well plate containing collagen-coated coverslips (density = $5 \times 10^4$) and incubated at 37 °C with 5% $CO_2$. Medium was refreshed after 4 h of seeding. After 24 h of seeding, medium was replaced by 199 medium containing 10 mM HEPES (Gibco), 4.5 mg/ml glucose, 2 mM L-glutamine (Invitrogen), antibiotic-antimycotic (Gibco), 20 mM sodium-pyruvate (Gibco), 5 ng/ml sodium selenite (Sigma), 5 mM nicotinamide, 10 mg/ml transferrin (Sigma), 10 mM 3,3′,5-triiodo-L-thyronine sodium salt (Sigma), 50 ng/ml recombinant mouse EGF (Invitrogen), 100 nM

**Fig. 7 AZD1152 inhibits hyperpolyploidization of hepatocytes and preneoplastic foci formation in DEN-treated liver. a** Immunohistochemistry of liver sections from indicated mice. The high magnification shows the detailed information of BrdU-positive hepatocytes. Schema of pharmacological manipulations was presented in Supplementary Fig. 7a. **b** The numbers of BrdU-positive hepatocyte and non-hepatocytes in indicated liver tissues with 1 month of drugs treatment (DEN, 50 mg/kg; AZD1152, 25 mg/kg or 50 mg/kg). N = 5 mice/group, and six different fields were selected randomly from each mouse. **c** Representative images from indicated liver tissues with 2 months of drugs treatment (DEN, 50 mg/kg; AZD1152, 25 mg/kg). The higher magnification shows the detailed nucleus morphology nearby CL and PL regions. **d** The nucleus size of hepatocytes within specific liver lobule in indicated mice from (**c**). N = 5 mice/group, and n = 750 hepatocytes/group. **e** Nucleus size of hepatocytes along CV-PV axis in indicated mice. N = 5 mice/group, and n = 750 hepatocytes/group. **f** The numbers of nucleus with vacuolation and vacuolation size in indicated liver after 2 and 3 months of drugs treatment (DEN, 50 mg/kg; AZD1152, 25 mg/kg). Twenty different regions were quantified from 5 mice for each group. **g** The numbers of preneoplastic foci in indicated liver after 3 months of DEN treatment. N = 5 mice/group. **h** Representative livers from control (Ctrl), DEN treatment (DEN), DEN treatment followed by 2 months AZD1152 (DEN + AZD) injection were sampled 6 months after DEN treatment for tumor size and numbers examination. **i** Quantitative data show the tumor diameter and tumor numbers in (**f**). N = 5 mice/group. Statistic: One-way ANOVA (**b**), (**d**), (**f**), (**g**), and (**i**); two-way ANOVA (**e**). Values represent the mean ± SEM. n.d., not detectable. Scale bars: 10 mm in (**h**) (also see Supplementary Fig. 7).

insulin (Sigma), 100 nM dexamethasone (Sigma), and 5% FBS. Medium was refreshed every 24 h throughout the culture time.

**Time-lapse recording**. The cytokinetic structures of hepatocytes were monitored in vitro by time-lapse microscopy (Leica, DMI 6000B) equipped with auto-focus system (MAC 6000 system) and live-cell instrument (CU-109, $CO_2$/air gas mixer FC-5). MetaMorph (version 7.6.5.0) was conducted for machine operation and image acquisition. Primary cultured hepatocytes isolated from the livers of 3-week-old mice were used since the proliferation ability of hepatocytes at this age is most vigorous and over 90% of hepatocytes are diploid and tetraploid[13]. Twenty-four hours after seeding, images of hepatocytes were recorded and cellular behavior and structures were visualized by phase-contrast system with HC PL Fluotar 20×/0.50 PH2 dry objective lens and 100 ms exposure time. Hepatocytes were incubated in a chamber maintained at 37 °C and 5% $CO_2$. Images were taken at 10-min intervals for ~150 h by Andor iXon EMCCD Camera. The optimal focal plane was set at beginning of each image session and adjusted by auto-focus system with 5-s intervals throughout the image recording. The recorded images were further analyzed via ImageJ software.

**Isolation of centrilobular and perilobular hepatocytes**. Enrichment of hepatocytes from CL or PL region was performed by using the digitonin-collagenase perfusion system as previously described[36,37]. Mice were anesthetized with Avertin (0.25 mg/g body weight) before liver perfusion. For isolation of centrilobular hepatocytes, 24G i.v. catheters (Terumo) were set into portal vein and inferior vena cava, and blood was removed by perfusion with $Ca^{2+}$- and $Mg^{2+}$-free HBSS from portal vein with a flow rate of 5 ml per min for 3 min at RT. Digitonin buffer (4 mg/ml in $Ca^{2+}$- and $Mg^{2+}$-free HBSS) was infused at a rate of 10 ml/min at RT from portal vein until a regularly scattered periportal discoloration pattern emerging on the surface of liver (this took 10–30 s). Digitonin was then removed by retrograde perfusion with $Ca^{2+}$- and $Mg^{2+}$-free HBSS containing 1 mM EGTA from inferior vena cava for 2 min with a flow rate of 5 ml/min. Retrograde infusion of 1 mg/ml collagenase I in $Ca^{2+}$- and $Mg^{2+}$-positive HBSS from inferior vena cava with a flow rate of 3 ml per min for 10 min was performed subsequently. After 10 min of collagenase perfusion, moved digested liver into DMEM medium, and liver was disrupted with scissors followed by dissociation of hepatocytes with pipetting. Cell suspension was filtered through a cell strainer (100 μm²) and mixed with equal volume of 90% Percoll (Sigma) in HBSS to collect live centrilobular hepatocytes. For isolation of perilobular hepatocytes, blood was removed by perfusion with $Ca^{2+}$- and $Mg^{2+}$-free HBSS from central vein followed by digitonin buffer infusion. Digitonin was then removed by retrograde perfusion with $Ca^{2+}$- and $Mg^{2+}$-free HBSS containing 1 mM EGTA from portal vein, which is followed by infusion of 1 mg/ml collagenase I in $Ca^{2+}$ and $Mg^{2+}$-positive HBSS from portal vein for 10 min. Subsequently, live perilobular hepatocytes were isolated by Percoll-dependent centrifugation method for further inoculation.

**Tumorsphere assay and colony formation assay**. Centrilobular and perilobular hepatocytes were isolated by the digitonin-collagenase perfusion system from control and 1 month DEN-treated mice. Cells were resuspended in 199 culture medium containing 10 mM HEPES, 4.5 mg/ml glucose, 2 mM L-glutamine, anti-biotic-antimycotic, 20 mM sodium-pyruvate, 5 ng/ml sodium selenite, 5 mM nicotinamide, 10 mg/ml transferrin, 100 nM insulin, 100 nM dexamethasone, 10 mM 3,3′,5-triiodo-L-thyronine sodium salt, 50 ng/ml recombinant mouse EGF, 20 ng/ml bFGF, 2% B27 supplements (Gibco), and 1% $N_2$ supplement (Gibco). Cells were then cultured in Ultra-low non-attachment 24-wells plate (Corning, 3473) with a serial dilution including 500, 1000, and 2000 cell number at 37 °C and with 5% $CO_2$. The medium was refreshed every week to prevent disturbance the formation of the tumorspheres. After 1-week incubation, pipetting was performed gently with 1 ml tips to dissociate aggregated cells, and tumorspheres were monitored by phase-contrast microscope (Leica, DMI 6000B) with HC PL Fluotar

20×/0.50 PH2 dry objective lens. Numbers of tumorsphere (diameter >100 μm) were analyzed by ImageJ. For colony formation assay, Ultra-low non-attachment plate was replaced by regular 6-well plate (Greiner CELLSTAR, 657160) with Type I collagen coating. After 21 days inoculation, cells were washed with PBS and then fixed by 4% PFA in PBS for 1 h at RT, which is followed by crystal violet staining (0.5% w/v) and counted using a stereomicroscope.

**Subcutaneous implantation**. Hepatocyte spheres derived from CL or PL hepatocytes were conducted for subcutaneous injection. Spheres derived from 2000 numbers of hepatocytes were used for implantation. During the subcutaneous implantation, spheres in 100 μl 50% Matrigel/media without serum was injected into subcutaneous region of skin. Spheres derived from PL- and CL-enriched hepatocytes were injected into the left and right flank under the skin of the dorsal body surface of each mouse, respectively. Mice were monitored daily in the first 10 days of post-injection and when the tumor started to grow, the tumor size was measured every 5 days. Tumor size was measured using a caliper. When the tumor size reached around 1 cm³, mice were euthanized and the tumor was harvested.

**Definition of preneoplastic lesion**. The sequentially pathological changes have been well described in human and rodents, and the terminology and criteria for various lesions have been defined based on consensus diagnosis[31,32]. The first visible pathological lesion in the progression of cancer can be classified as a pre-neoplastic lesion, characterized by as a small focus of cellular alternation and appear to represent clonal expansion of about a hundred cells. These preneoplastic lesions are difficult to detect with the naked eyes but can be identified microscopically. For tumor nodules, constituted by continuously growing of pre-neoplastic cells, can be observed directly through naked eyes on the tissue surface. Tumor nodule may identify as benign tumor since they grow by expansion of preneoplastic cells and are not invasive with well-differentiated morphology. For instance, in Fig. 1b, the whitish and round clump on the liver surface can be defined as a tumor nodule.

**Immunohistochemistry and H&E staining**. After anesthetization, mice were perfused with cold PBS followed by 4% paraformaldehyde (PFA)/PBS. Specimens were post-fixed with 4% PFA/PBS and embedded in paraffin-wax with standard protocol. Five-micrometer-thick paraffin sections were prepared as 10 μm thickness and deparaffinized with xylene. Rehydration was performed with a serial concentration ethanol followed by antigen retrieval in Tris–EDTA buffer (pH 9.0) for 5 min with pressure cooker. Sections were immersed in PBS containing 0.2% Triton X-100 for 30 min at RT for permeablization. For immunofluorescence labeling, sections were treated with Super Block and mouse to mouse block (ScyTek Lab) for 10 min and 60 min followed by blocking with 5% BSA and 5% FBS in PBS for 1 h. After blocking, sections were treated with primary antibodies diluted in PBS containing 1% BSA, 1% FBS, and 0.05% Triton X-100. Following overnight incubations at 4 °C, sections were washed extensively with PBS containing 0.1% Triton X-100 and then incubated with secondary antibodies conjugated with Alexa fluorophores and DAPI. After PBS wash, sections were mounted in Prolong Gold mounting medium with anti-fade reagent (Invitrogen) for image acquisition. Histological staining with hematoxylin and eosin was performed by the institutional Pathology Core staff. Sections after the removal of paraffin and rehydration were stained with hematoxylin and eosin according H&E staining standard protocol.

**Immunocytochemistry**. Hepatocytes inoculated on collagen-coated coverslips were washed with PBS and then fixed with 4% PFA/PBS at RT for 10 min. Permeablization was performed with 0.2% TX-100 in PBS at 4 °C for 20 min. Hepatocytes were then conducted for blocking with 5% BSA and 5% FBS in PBS for 1 h at RT. For immunofluorescence labeling, hepatocytes were treated with primary antibodies diluted in PBS containing 1% BSA, 1% FBS, and 0.05% Triton X-100.

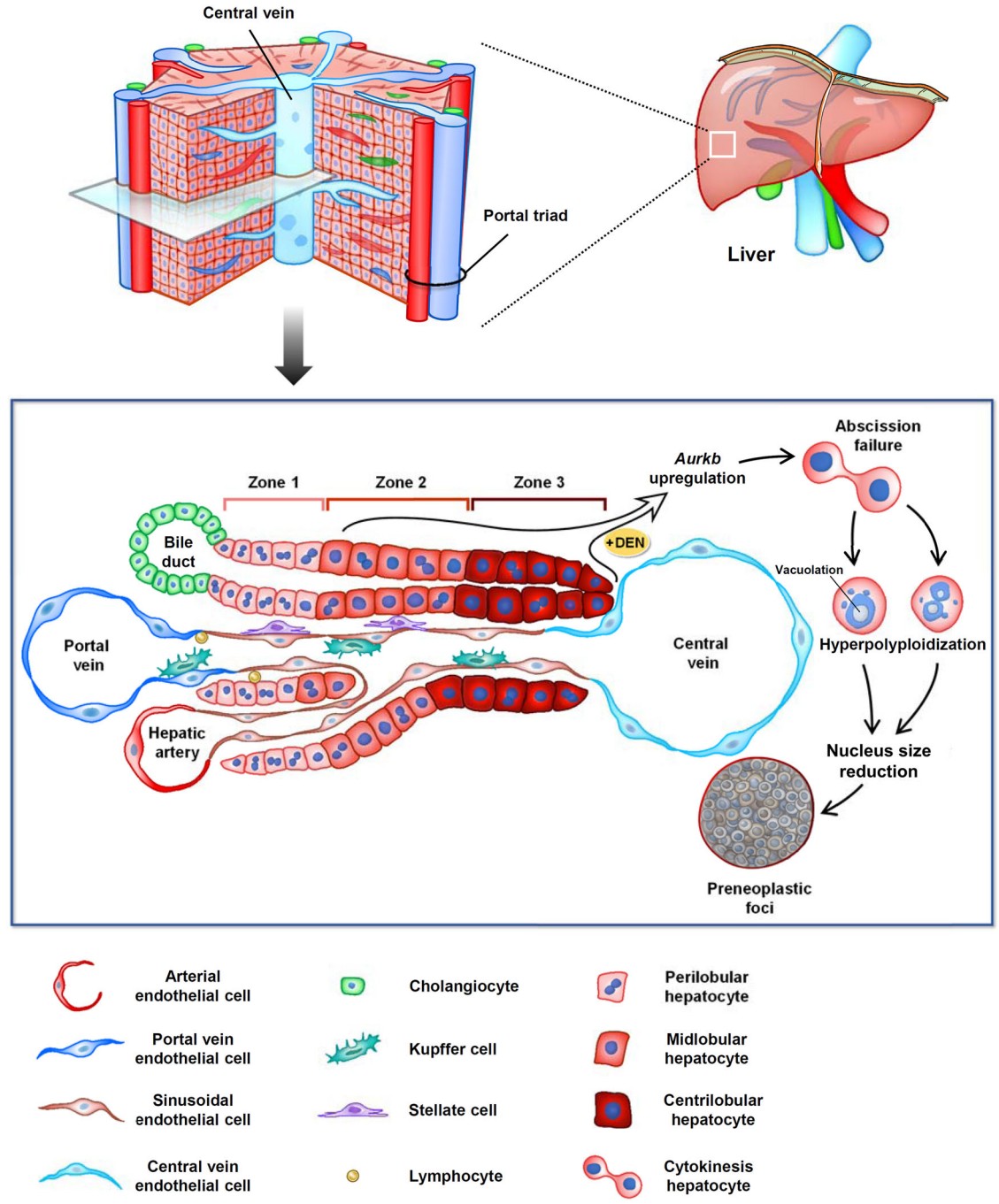

**Fig. 8 Schematic illustration of the generation of preneoplastic lesion from hyperpolyploid hepatocytes.** Metabolically, hepatic lobule is composed by acinus that can be classified into three Zones: Zone 1 perilobule (PL), Zone 2 midlobule (ML), and Zone 3 centrilobule (CL). DEN treatment causes upregulation of *Aurkb* and abscission failure, which is a critical step for hyperpolyploidization of hepatocytes. The ratio of hyperpolyploid hepatocytes is increased predominantly in CL and ML region after DEN injection, accompanied by nuclear vacuolation and micronuclei. Hyperpolyploid hepatocytes might be the cell of origin in tumor initiation and subsequently transform into small size preneoplastic cells via an unknown process.

Following overnight incubations at 4 °C, hepatocytes were washed extensively with PBS containing 0.05% Triton X-100 and then incubated with secondary antibodies conjugated with Alexa fluorophores and DAPI. After PBS wash, coverslips were mounted in Prolong Gold mounting medium with anti-fade reagent (Invitrogen) for image acquisition.

**Image acquisition**. For evaluation of nuclear and cellular size in the liver, DAPI labeling and anti-β-catenin antibody staining were applied to outline nuclear and cellular architecture, respectively. Immunostained tissues were observed with a confocal laser scanning microscope LSM780 (CarlZeiss, Germany) equipped with an argon laser (excitation 488 nm) and a DPS laser (excitation 561 nm). Serial optical Z-sections were acquired using a Plan Apochromat 10X/0.45 M27 objective with 2048 × 2048 resolution. The x–y plane resolution as calculated by scaling 0.629 μm with apertures 29 μm. For observation of signals at midbody region, serial optical Z-sections were acquired using a Plan Apochromat 20X/0.8 M27 objective with 2048 × 2048 resolution. The x–y plane resolution as calculated by scaling 0.231 μm with apertures 27 μm. The nuclear budding of hepatocytes was detected by serial optical Z-sections using a Plan Apochromat 40X/1.4 Oil DIC M27 objective with 2048 × 2048 resolution. The x–y plane resolution as calculated by scaling 0.148 μm with apertures 29 μm. Projection images were generated using ZEN black image analysis software (CarlZeiss, 2011 SP7 FP3, version 14.0.0.0) with full resolution and shown for architecture of hepatocytes in vivo and in vitro.

**Images quantification**. For the nuclear and cellular size measurement alone CV-PV axis, the CV-PV axis was subdivided into 15 parts (Supplementary Fig. 1a), and the less number indicates the region closed to central vein region[13]. The traditional 3-zone classification of hepatic lobule divides CV-PV axis into three equal parts: namely, centrilobular zone corresponds to 1–5, midlobular zone to 6–10, and periportal zone to 11–15. Only hepatocytes (cells with round or oval nuclei) were analyzed in vivo and in vitro. Nuclear and cellular sizes were measured according to DAPI and β-catenin signal, respectively. Data were obtained from three to five mice for each group. For quantification, at least five different fields of the microscope were selected randomly from each mouse liver, and ~50 CV-PV axises were analyzed for quantification. Mono- and bi-nucleated hepatocytes were distinguished by comparing DAPI and membrane-labeling images.

The length of intercellular bridge between daughter hepatocytes was quantified based on the β-tubulin signal[13]. Intercellular bridge is a region with highly condensed microtubules, and it is defined as a transient structure with about 1–2 μm in diameter only appearing toward the end of cytokinesis and just prior to the complete separation of nascent daughter cells. Length of the brightly labeled tubulin bundle in the center of the intercellular bridge between nascent daughter cells was measured using ImageJ. The subcellular fluorescent intensity and length of AURKB and pT232-AURKB during abscission were quantified by background-corrected line scans along the central spindle region or midbody using ImageJ. The integrated fluorescence intensity along this line was used as the estimated amount of specific protein in this region. Over 100 dividing hepatocytes were analyzed for each group from three independent experiments. All images were acquired with a fixed exposure time and condition in the same experiment. All quantification of fluorescence intensities was performed under raw 16-bit images, and statistical analysis was performed by GraphPad Prism version 8.4.0 (671).

For quantification of the size and numbers of budding nuclei and micronuclei in cultured hepatocytes, DAPI and β-tubulin signals were conducted to outline nuclear and cellular morphology of hepatocytes, respectively. Nuclear budding was defined as a protrusion part from regular nuclei, and micronuclei was characterized as a small separated part with DAPI positive signal. The size and numbers of budding nuclei and micronuclei were analyzed by ImageJ. Over 150 cultured hepatocytes with nuclear budding and micronuclei were analyzed for each group from at least three independent experiments.

Quantitative analysis of cell division types from time-lapse images of hepatocytes in culture was performed by ImageJ. The complete cytokinesis, cytokinesis failure, and abscission failure are defined as previous described[13]. Complete cytokinesis is defined as two individual daughter cells generated from a single hepatocyte after bipolar dividing. For cytokinesis failure, hepatocytes perform nuclear division (karyokinesis) only, and no contractile ring formation and cytokinesis are progressed. For abscission failure, ingressed cleavage furrow emerges between two daughter cells; however, the daughter cells lose its ability to cut off intercellular bridge, and furrow regression is performed to generate a single cell with multiple nuclei.

**Flow cytometry**. Hepatocytes were isolated by collagenase perfusion system and resuspended in ice-cold PBS at a density of $2 \times 10^6$ cells/ml. Cells were fixed with 4% PFA at 4 °C overnight with gentle rotation. Cells were washed with PBS containing 0.25 mg/ml RNase A and stained with 2 μg/ml DAPI at 4 °C for 60 min. DNA content of labeled cells was measured by flow cytometer (Becton Dickinson, LSRII SORP – 17 color analyzer). Acquired data were analyzed by BD FACSDiva software v6.2 and FlowJo 7.6.1, Engine 2.79000 OS version.

**Quantitative real-time PCR**. Total RNA in liver was purified by TOOLSmart RNA Extractor (BioTools, DPT-BD24) and RNeasy Mini kit (QIAGEN, 74104). The isolated RNAs were reverse transcribed by using an oligo-dT/random primer mixture and ImPromII Reverse Transcriptase (Promega) with an equal amount of purified RNA. Quantitative(q) PCR involved the Universal Probe Library and Lightcycler 480 system (Roche). The comparative Ct (threshold cycle value) method was used to calculate relative expression. Raw data were normalized by the expression level of *Actb*, which showed no significant difference in both control and drug-treated mice. The primer pairs used in qPCR are listed in Supplementary Table 1.

**Immunoblot assay**. Protein extracts were prepared from equal amounts of the liver or cultured hepatocytes in Laemmli sample buffer supplemented with 1X protease inhibitor cocktail (Roche) and phosphatase inhibitors (Tocris). Homogenates were sonicated by ultrasonic cell disruptor (Misonix Sonicator 3000) with power level 2 to disrupt genomic DNA, and insoluble debris was removed by centrifugation at $14,000 \times g$ and 4 °C for 10 min. Protein lysates were boiled at 55 °C for 15 min and then resolved by 10% SDS-PAGE mini-gels, transferred to nitrocellulose membranes. The membranes were then blocked in 5% skim milk in 1X TBST (TBS with 0.1% Tween 20) for 60 min, followed by the incubation of indicated primary antibodies at 4 °C overnight. Specific horseradish peroxidase-conjugated secondary antibodies were used for enhanced chemiluminescence detection (ECL-Prime, GE Healthcare Life Sciences) by Image Quant LAS 4000 (Fujifilm).

**Statistical analysis**. Mice and cultured hepatocytes were randomly assigned for time-course study and drugs treatment. Imaging fields were randomly selected during image acquisition. The sample size among experimental groups was kept as equally as possible. The experiments and analysis were conducted in a blind manner and replicated at least three times independently. For the in vivo studies, at least five mice were used for each group. GraphPad Prism version 8.4.0 (671) software was applied to produce the graphs and statistical analysis.

We conducted experiments including BrdU-positive nuclear area (Supplementary Fig. 1j), nuclear size between vacuolation and non-vacuolation hepatocytes (Fig. 2b), nuclear-to-cytoplasmic ratio (Fig. 2h), nuclear and cytoplasmic size between tumor and non-tumor cells (Fig. 3a), nuclear size of colony and non-colony cells (Fig. 3i), nuclear size of region-specific hepatocytes (Supplementary Fig. 3b), tumorsphere size (Supplementary Fig. 3f), *Aurkb* gene expression in HCC patients (Fig. 4f) to two-tailed unpaired Student's *t*-test with Welch correction. Supplementary Fig. 4c was analyzed statistically by two-tailed paired Student's *t*-test. For the ratio of different cell division type (Fig. 1i), qPCR analysis of CL and PL markers expression (Fig. 3e), and expression of cytokinesis genes (Fig. 4c), two-tailed Mann–Whitney nonparametric test was applied.

For the effect of time on nuclear, cellular and micronuclear size after drugs treatment (Figs. 1d, 6b, 7d, Supplementary Fig. 1f, k, dot plot graphs of Supplementary Fig. 6c, d, and Supplementary Fig. 7c), numbers of BrdU-positive hepatocytes (Fig. 1f), distribution of vacuolation nuclei and tumor foci (Fig. 2d and i), colony numbers (Fig. 3g), *Aurkb* gene expression in various liver disease patients (Fig. 4e), the intensity of pT232-AURKB and AURKB signals at the midbody (Fig. 5c), the length of intercellular bridge, pT232-AURKB and AURKB signals (Fig. 5d, Supplementary Fig. 5c), the ratio of different cell division type after drugs treatment (Fig. 5e), the change of nuclear size after drugs treatment (Fig. 6b), the effect of AZD1152 on the numbers of BrdU-positive cells, nuclear size of hepatocytes, numbers and size of vacuolation in hepatocytes, and preneoplastic foci numbers (Fig. 7b, d, f, g), the numbers of EpCAM and Vimentin double positive cells (Supplementary Fig. 6g), and the tumor diameter and tumor numbers in Fig. 7f, g, i were statistically analyzed by one-way ANOVA.

Two-way ANOVA was applied to analyze the distribution of BrdU-positive cell along CV-PV axis at different stages of DEN treatment (Fig. 1g and Supplementary Fig. 1i), the effect of DEN on the numbers of vacuolation and preneoplastic foci at different time-courses (Fig. 2c, e), in vivo tumor formation rate (Fig. 3j), the effect of region-specific hepatocytes on tumorsphere formation (Supplementary Fig. 3e), the effect of AZD1152 on the size of nuclei and micronuclei, percentage of cells with micronuclei and nuclear budding, and the numbers of micronuclei in a cell at different time points (Fig. 6c, e, and f, and the line graphs of Supplementary Fig. 6c, d), and the effect of drugs on the frequency distribution of nucleus and cell size along CV-PV axis (Fig. 7e, and Supplementary Fig. 7d). Bonferroni's multiple comparison test was applied for comparisons among multiple conditions following one-way or two-way ANOVA tests.

Correlation quantifies the degree to which two variables are related including expression level of *Aurkb* and the survival time and tumor doubling time of patients (Fig. 4h and Supplementary Fig. 4d), nuclear size, the numbers and size of vacuolation (Supplementary Fig. 2c), the length of intercellular bridge and AURKB intensity (Supplementary Fig. 5d) were statistically analyzed by correlation coefficient. Kaplan–Meier survival analysis was conducted to statistically analyze the overall survival curves of patients with different *Aurkb* expression (Fig. 4g). Sample numbers and statistical results were indicated in the figure legends precisely. Data were presented as the mean ± standard error (SEM.) n.d., not detectable.

**Reporting summary**. Further information on research design is available in the Nature Research Reporting Summary linked to this article.

## Data availability

Data supporting the findings of this study are available within the article and its supplementary information files and from the corresponding author upon reasonable request. A reporting summary and Supplementary Figures for this article are available as a Supplementary Information file. The NCBI GEO datasets used in this study are as follows: GSE19057, GSE63726, GSE89632, GSE66232, GSE20140, GSE54236, and GSE64041. Source data are provided with this paper.

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

## Acknowledgements

The authors are indebted to Dr. Yi-Shuian Huang and Dr. Ruey-Bing Yang for supplying facilities and drugs for in vivo and in vitro studies. The authors thank the pathology core of Institute of Biomedical Sciences (IBMS) for paraffin blocks and H&E stain processing. The authors also thank the assistance from Ms. Tzu-Wen Tai and Ms. Chia-Chen Tai in the flow cytometry core facility of IBMS (AS-CFII108-113), Academia Sinica for analysis of flow cytometry. The authors are grateful to Mr. Kuan-Yu Chou and Ms. Show-Rong Ma in microscopes facility of IBMS for images acquisition. We thank Mr. Tsung-Hung Hung in Medical Art Room of IBMS for medical art construction. This work was supported by grants from the Taipei Medical University [105-5406-004-112], Ministry of Science and Technology of Taiwan [MoST108-2628-B-038-002 and MoST106-2320-B-038-026].

## Author contributions

Experimental conception and design: H.L., Y.-S.H., J.-M.F., M.D., H.C., and H.-W.C.; development of methodology: H.L., Y.-S.H., J.-M.F., M.D., H.C., H.-S.H., and H.-W.C.; acquisition of data: Y.-S.H., H.H. Lai, S.-H.L., Y.-L.L., P.-C.K., H.-S.H., H.-W.C., and P.-Y.Y.; analysis and interpretation of data: H.L., Y.-S.H., J.-M.F., M.D., H.C., S.-H.L., and Y.-L.L.; drafting of manuscript: H.L., Y.-S.H., and H.-W.C.; critical revision: H.L., Y.-S.H., J.-M.F., M.D., H.C., and H.-W.C.; study supervision: H.-W.C.; Jean-Michel Fustin and Masao Doi share co-second authorship.

## Competing interests

The authors declare no competing interests.
