## [Peer Review File · Nature Communications]

Reviewers' comments:

Reviewer #1 (Remarks to the Author):

The manuscript by Lin and colleagues provides an important analysis of HCC formation, uncovers an interesting mechanism driving tumorigenesis and identifies a novel therapy. First, they analyze livers from DEN-injected mice at relatively early time points, 1-3 months post injection, when only preneoplastic or small tumor lesions are formed. They find evidence of large hepatocytes with large nuclei (hyperpolyploid cells) localized mostly to the centrilobular region (CL), which likely formed through defective cytokinesis. Cells within the CL region express proliferation markers and many may be damaged. Consistent with cellular changes in the CL region, tumor lesions are shown to originate in the CL region. In contrast to the hyperpolyploid hepatocytes that arise early, the tumor lesions are comprised of small cells with immature characteristics, and the authors suggest that genome reductions by hyperpolyploids are the source of these tumorigenic cells (I have major issues with this; see below). Secondly, the authors identify AURKB as a putative cytokinesis regulator upregulated in DEN-induced HCC and in human HCC. AURKB inhibition with a small molecule reduces hyperpolyploidy and ameliorates cellular changes and tumor foci associated with DEN treatment. Overall, the experiments are performed well, statistics are appropriate and the writing is solid. The finding that DEN-treatment leads to hyperpolyploid hepatocytes is entirely novel. Demonstration that AURKB inhibition attenuates HCC provides an important protein that could be targeted with existing drugs. There are a number of concerns that must be addressed, and these are detailed as Major and Minor concerns below. My primary issue involves the link between hyperpolyploid hepatocytes and the small, immature hepatocytes that comprise the tumors. Their data do not support the model described.

MAJOR CONCERNS

1. A major finding of the paper is that DEN treatment leads to an enrichment of hyperpolyploid hepatocytes in the CL region of the liver. Data suggest that these hyperpolyploid hepatocytes arise by abscission failure, which could be attributed to upregulated expression and activity of AURKB. To what extent are these findings applicable to HCC in general? For example, I suspect the reason hepatocytes in the CL (and sometimes ML) region are affected is because the enzyme involved in DEN metabolism, CYP2E1, is expressed in the CL region, which makes these cells especially sensitive to DEN treatment (Cancer Research 2007; 67: 11141). My basic question is this – are the findings simply an artifact of the DEN model, or are they relevant to the early-stage HCC formation? To address this, I would recommend demonstrating the findings (at least a minimum set of core observations) in an alternate tumor model. This would greatly strengthen the relevance of the work.

2. The authors show in Fig. 3a that tumor foci are enriched with small cells with small nuclei. They speculate that the founding hyperpolyploid hepatocytes in these nodules undergo ploidy reduction, and it is these cells that expand/proliferate in the tumor. Unfortunately, the experiments in Fig. 3, described on pages 12-13, do not adequately test this idea. The digitonin perfusion system is used to isolate CL- or PL-enriched hepatocytes (this is a nice use of a classic technique!) from DEN-treated mice (how long after DEN?). The CL fraction is enriched with polyploid hepatocytes and CL gene signatures, and the PL fraction is enriched with diploid hepatocytes and PL gene signatures. Using two different *in vitro* tumor assays, the CL fraction is shown to produce more/bigger colonies, which is interpreted as having more tumor activity. I have several concerns. First, demonstration of tumor features by a specific cell population using *in vitro* assays does not necessarily reflect the *in vivo* situation, which in my opinion makes it hard to conclude anything. Secondly, even if the *in vitro* approach were valid, an important control is missing. The *in vitro* studies are done with hepatocytes from DEN-treated livers, but not control livers. What would happen if CL and PL hepatocytes from untreated livers were compared in the same assays? Thirdly, the experiments never consider whether enrichment of big, polyploid hepatocytes in the

CL region is independent of the small hepatocytes that expand to comprise tumor lesions. These questions need to be addressed.

3. Fig. 6d-f and associated supporting figure. It is interesting that DEN treatment leads to the formation of micronuclei and nuclear budding and that these effects are ameliorated by AZD. However, it is a conceptual leap to imply this is a mechanism of genome reduction. Consider the following examples:

* Page 17 (line 332): "Together with our finding presented in Fig. 3, these results paint a picture in which aberrant AURKB-induced hyperpolyploidy plays a role in the transformation of CL-hepatocytes into preneoplastic cells via genome reduction."

* Page 22 (line 427): "According to colony formation assay, our findings support the hypothesis that CL-hepatocytes may be the origin of HCC cells through genomic content reduction²²."

* Fig. 8. The part about "genomic content reduction" is misleading.

There is no evidence that genome reduction is happening or that any particular process is driving genome reduction. Direct evidence must be presented, or the entire model needs to be revised. It would be a lot of work, but it might be possible to demonstrate ploidy reduction using the elegant lineage tracing model that was recently described by Matsumoto and Grompe (*Cell Stem Cell* 2020;26:34).

4. It is shown in Fig. 2 that hepatocytes with vacuoles are enriched in the CL region, and this is interpreted as an indicator of senescence and precancerous phenotype. This is fairly basic characterization. What else is happening to the cells? For example, do they have other features of senescent cells (e.g., beta-gal expression, shortened telomeres); do they have damaged DNA or oxidative damage? A more detailed characterization is required.

MINOR CONCERNS

1. Work from Bou-Nader, et al. (Ref 21) is mentioned in the Introduction (page 6; line 111). The paper shows that human HCC are characterized by increased nuclear polyploidy. This is an appropriate paper to cite, but it is important to emphasize that the tumors with increased polyploidy have mutations in p53. This should be added to the text.

2. Page 6 (lines 114-116). What type(s) of "hyperpolyploid giant polyploid" cells are being described here? Without mentioning the cell or tissue type, the next sentence doesn't make sense.

3. The designation of ploidy types in Supplementary Fig 1b is not accurate. The math at the bottom makes sense assuming the cells are in the same phase of the cell cycle (e.g., in G0/G1). However, if some cells have replicated their DNA and some have not, then the scaling does not make sense. I suggest using "chromatid number" (c) instead of "chromosome number" (n). For instance, in the case of a cell with tetraploid content, it could be either a tetraploid cell in G0/G1 (i.e., 4n4c) or a diploid in G2/M (i.e., 2n4c). To avoid confusion, it is safest to call the population 4c. I suggest labeling the populations in Supporting Figure 1b as 2c, 4c, 8c, etc. Ploidy populations in the FACS plots should be renamed as well.

4. Live cell imaging is used to illustrate patterns of cell division by hepatocytes from control and DEN-treated mice. One can generally appreciate the patterns of cytokinesis described in the text, but it is very difficult to see cell boundaries and to appreciate the outcome of each division. It is especially difficult to appreciate abscission failure. Technical limitations of imaging include the following: (a) use of phase contrast only (can markers be used to highlight microtubules or cell membranes?); (b) cells are packed densely; (c) duration of imaging is limited. For example, in Supplementary Video 4, one can see the first-round abscission failure, but the second-round failure is not convincing because the movie ends so quickly. In Supplementary Video 5, it is hard to appreciate that cytokinesis is complete. I fully recognize that live cell imaging of primary hepatocytes is challenging, but the data would be much more convincing with clearer, less

ambiguous movies. Additionally, it would be helpful to add time stamps to the movies.

5. Tumor nodules at 1, 2 and 3 months post DEN injection are continuously classified as “preneoplastic” lesions. What criteria are the authors using to distinguish between “preneoplastic” and cancerous lesions. For example, is the small nodule in Fig. 1b preneoplastic or is it a tumor? The authors should describe their classification system.

6. The writing is overall very good, but there are a number instances throughout the text where an incorrect word is used. These issues should be addressed.

Reviewer #2 (Remarks to the Author):

The manuscript from Lin et al. entitled ‘Hyperpolyploidization of hepatocyte initiates preneoplastic lesion formation in the liver’ aims to address an important question regarding the early stages of hepatocellular carcinoma development and the role of polyploidization, and mechanistically the role AURKB plays, in the formation of this tumour. Hepatocellular carcinoma (HCC) is a major unmet clinical need worldwide and there is an enormous requirement to understand the transition of healthy precursor hepatocytes into pre-neoplastic clones and then onto cancer itself. Dissection of this axis will be critical for preventing and early detection of this disease.

To answer this question the group use a mouse model and comparison to published human datasets from NAFLD, cirrhosis and HCC. The mouse model, DEN, is appropriate but as a model system it is used in isolation and there are significant caveats to how well it models both the progression of human HCC and the genetic landscape of HCC (Connor et al. <https://doi.org/10.1016/j.jhep.2018.06.009>). Using intensive analysis of IHC, combined with cell culture and mechanistic testing using the AZD1152 inhibitor the author show that in this model that hepatocytes enlarge and become polyploid in a zonal manner post DEN. Using in vitro analysis they show this is related to incomplete cytokinesis and abscission failure and that ‘transformed’ hepatocytes from the zones with highest polyploidy are most likely to be clonogenic. After identifying AURKB from published data sets they go onto to test the specific AURKB inhibitor AZD1152 in vitro and in vivo and relate changes to impaired abscission and formations of ‘preneoplastic’ foci.

The authors are to be congratulated on this large body of work and the development of additional technical advances including the development of techniques for zonal isolation of hepatocytes. In general the major claim of the paper that hyperpolyploidy is induced by DEN and through AURKB mediated failure of appropriate cell division leads to the development of preneoplastic foci in this model of HCC. These findings are novel and provocative and of interest to others in the community, but of less interest to the wider field. The findings will influence thinking in the field however I think that significant caveats to the model system used should be highlighted not to overstate the certainty of this process in the human. In general the work would be reproducible given the level of detail provided however in some instances (as detailed) additional methodological description should be provided.

Major comments

Central limitations of this study are:

1. The authors show an altered behaviour of polyploid hepatocytes after DEN, and reduced ‘preneoplastic;’ burden after AZD1152. We do not know however which of the many abnormal nodules which form in DEN go onto tumours and how representative they are of human disease progression, especially in the face of cirrhosis. The authors do not show that either the polyploid hepatocytes or the smaller preneoplastic foci are genuine precursors of cancer in this model (transplant experiments could demonstrate this), or that endstage tumour/disease burden is altered upon early AZD1152 treatment. Showing either of both of this would significantly increase the impact of the paper.
2. The relevance of the DEN mouse model to human HCC. This could be mitigated by alternative

HCC models in the mouse. This discussion of accuracy of animal models is much debated in the field currently and there is no simple solution experimentally currently. I would recommend this point is discussed by the reviewers in greater detail but I do not see that repetition of this work in additional model systems if required for this current body of work. The major changes of DEN resulting in nuclear enlargement and polyploidy in the centrilobular area may well be a reflection of this being the area of principle cellular damaged after acute DEN administration. This should be discussed, but the data on reversal of ploidy and reduction of 'preneoplastic foci' with AURKB inhibition reassures that at least ploidy is related either directly or indirectly to foci formation.

3. If polyploidy hepatocytes are the source of HCC then mechanisms of genetic reduction remains unanswered. The authors highlight this appropriately but further work in this areas would be welcome in the future. Clearly a link between polyploidisation in early cancer and genetic-reduction in cancer formation would provide a significantly more exciting and complete study.

Additional points include:

The introduction discusses genetic lineage tracing studies from hepatocytes – it should be highlighted that these are in mouse models. The introduction should not leave the reader with the idea that in human other sources of HCC have been excluded.

The timepoint of non-DEN treated control should be stated in each figure. As polyploidy changes over time in the mouse liver it is important to compare non-DEN and DEN at matched timepoints throughout. From my understand the controls used are aged matched to p15+3M but this should be explicitly stated and ideally other aged match controls used when comparing the other timepoints. Similarly non-DEN control should be clearly identifiable in the key figure 5G.

Additional detail is required in the methods. In what vehicle was the AZD1152 delivered? Which stock of BrdU was used?

In general the figures are of good quality but the legends could make it easier to understand what was done in the experiments and what is being used as a control. Data in Figure 3G should be compared to hepatocytes from non-DEN treated animals. Statistical comparison of DEN and DEN & AZD1152 groups in Figure 5 should be shown. Is significant reversal upon AZD treatment observed?

Stylistic points

The authors switch between days and months post DEN, consistency here would help the reader. Please tone down strongly categorical descriptions of the data e.g. 'clearly indicates' change to 'indicates'. Similarly the complete lack of periportal polyploidization should not be excluded unless this is convincingly shown (lines 218 and 359). Additionally the comment 'until now' (line 416) referring to link to HCC should be nuanced – please place into context of my major comments 1 and 2. This also applies to the following sentence in line 418.

Labelling of tumour and non tumour is misleading e.g. Figure 2h.

It would be helpful to show increased DNA content in the enlarge hepatocytes after DEN earlier and ideally related to the other data in Figure 1.

Minor comments

Does dual inhibition of AURKA and AURKB have a similar effect; using ZM447439 in vitro for example. The addition of another pharmacological inhibitor would strengthen the study. This need not be completed throughout the experiments, but in some critical short term in vitro studies it would add further weight to the authors argument.

In our experience quantification of hepatocytes based solely on nuclei shape after immunofluorescence is challenging, larger leukocytes especially lymphocytes may also have larger nuclei and may be misclassified. The addition of a bona fida hepatocyte marker would help. Similarly whilst it is standard to quantify across the lobule from CV to PV the complex 3D nature of tissue may mean that zones are mis-defined. Large scale analysis of tissue helps to reduce this experimental 'noise' but ideally markers of zones could be used to define distinct areas rather than 2D morphology. Neither of these issues are necessary to address for revision but are mentioned as important to the field.

The zonal proliferation observed in Fig 1G is discrepant with that observed by others, notably that of Tchorz group. This does not require revision but can the authors speculate why they see CL

dominant proliferation in homeostasis compared to ML observed by others.

Line 409 – is this Figure 8?

Line 414 – the link to HCC disease progression is unclear from this sentence. Can this be amended for the clarity of the reader.

There are minor typographical errors in the main manuscript and figure legends which should be corrected.

Figure 3f – is this Hoeschst 33324?

**Point-by-point reply to Reviewers:**

**For reviewer #1:**

We are deeply honored by the time and effort you spent in reviewing this manuscript.
We have revised the manuscript thoroughly according to your suggestions. The
responses to your comments are showed point-by-point as follows.

*1. A major finding of the paper is that DEN treatment leads to an*
*enrichment hyperpolyloid hepatocytes in the CL region of the liver.*
*Data suggest that these hyperpolyloid hepatocytes arise by*
*abscission failure, which could be attributed to upregulated*
*expression and activity of AURKB. To what extent are these*
*findings applicable to HCC in general? For example, I suspect the*
*reason hepatocytes in the CL (and sometimes ML) region are*
*affected is because the enzyme involved in DEN metabolism,*
*CYP2E1, is expressed in the CL region, which makes these cells*
*especially sensitive to DEN treatment (Cancer Research*
*2007;67:11141). My basic question is this – are the findings simply*
*an artifact of the DEN model, or are the relevant to the early-stage*
*HCC formation? To address this, I would recommend*
*demonstrating the findings (at least a minimum set of core*
*observations) in an alternate tumor model. This would greatly*
*strengthen the relevance of the work.*

**Reply:**

We agree with reviewer's suggestion, and three different mouse models were applied
to address this question, including aflatoxin B1 (AFTB1) model [1], CCl₄ model [2], and
high-fat diet (HFD) model [3]. These three model systems were selected for the following
reasons. Firstly, AFTB1 is a common contaminant produced by *Aspergillus flavus* and *A.*
*parasiticus* in a variety of foods grains. It has been demonstrated as a very potent
carcinogen causing HCC in humans and rodents [4]. Secondly, CCl₄ one of the most
potent hepatotoxins causing HCC, was widely used as a cleaning fluid in the textile
industry, and in households as a spot remover for fabrics, which is another most potent
hepatotoxins caused HCC [5]. Finally, high-calorie diet caused nonalcoholic fatty liver

disease (NAFLD) is a widespread metabolic disorder that is reported as a risk factor for
HCC [6]. Additionally, high-fat diet induced hyperpolyploid hepatocytes via oxidative stress
in mouse has been demonstrated by Desdouets's group in 2015, but they did not specify
the distribution of these cells [3]. The results obtained with these three mouse models are
now showed in **Supplementary Fig. 1k**. We found that DEN-induced hyperpolyploid CL
hepatocytes obtained with DEN can also be observed in AFTB1-, CCl₄-, and HFD-treated
liver. As shown in **Supplementary Fig. 1k**, hepatocytes under these three conditions
displayed an enlarged nucleus size near the central vein region compared to control group.
These results indicate that hyperpolyploidization of hepatocytes around the central vein
region is not limited to DEN but appears to be a common event elicited by known
HCC-causing xenobiotics and oxidative stress such as AFTB1, CCl₄ and HFD.
Xenobiotics and oxidative stress being the major contributors to HCC, these new findings
strengthen the relevance of our study.

We added one paragraph describing the above findings in the manuscript,
accompanying the new **Supplementary Fig. 1k and I**, as follows.

(main text)

"To investigate whether hyperployploidization of CL hepatocytes is simply an artifact
of the DEN model or a relevant feature of early-stage HCC formation caused by
xenobiotics or oxidative stress, three different mouse models, using Aflatoxin B1 (AFTB1),
carbon tetrachloride (CCl₄), and 45 kcal% high-fat diet (HFD), were utilized to address
this question. As shown in Supplementary Fig. 1k, hepatocytes in AFTB1- and
CCl₄-treated mice displayed enlarged nucleus size near the central vein region, compared
to control group. After 90 days of HFD treatment, hepatocytes also exhibited slightly larger
nucleus size than age-matched control nearby central vein region, but not in the liver with
60 days of HFD treatment. These results indicated that not only DEN, but also xenobiotics
and HFD-induced oxidative stress specifically target CL and ML hepatocytes and cause
hepatic hyperpolyploidization within these two regions. More generally, the metabolic
sensitivity of CL and ML hepatocytes to xenobiotics is likely an important factor in HCC
development." (line 162-175)

2. The authors show in Fig. 3a that tumor foci are enriched with small cells with small nuclei. They speculate that the founding hyperpolyploid hepatocytes in these nodules undergo ploidy reduction, and it is these cells that expand/proliferate in the tumor. Unfortunately, the experiments in Fig. 3, described on pages 12-13, do not adequately test this idea. The digitonin perfusion system is used to isolate CL- or PL-enriched hepatocytes (this is a nice use of a classic technique!) from DEN-treated mice (how long after DEN?). The CL fraction is enriched with polyploid hepatocytes and CL genes signatures, and the PL fraction is enriched with diploid hepatocytes and PL gene signatures. Using two different in vitro

*tumor assays, the CL fraction is shown to produce more/bigger*
*colonies, which is interpreted as having more tumor activity. I have*
*several concerns. First, demonstration of tumor features by a*
*specific cell population using in vitro assays does not necessarily*
*reflect the in vivo situation, which in my opinion makes it hard to*
*conclude anything. Secondly, even if the in vitro approach were*
*valid, an important control is missing. The in vitro studies are done*
*with hepatocytes from DEN-treated livers, but not control livers.*
*What would happen if CL and PL hepatocytes from untreated livers*
*were compared in the same assays? Thirdly, the experiments never*
*consider whether enrichment of big, polyploid hepatocytes in the CL*
*region is independent of the small hepatocytes that expand to*
*comprise tumor lesions. These questions need to be addressed.*

**Reply:**

As the reviewer mentioned, *in vitro* assays do not necessarily reflect the situation *in*
*vivo*. To address the tumorous potency of CL and PL hepatocytes *in vivo*, subcutaneous
implantation of spheres derived from CL- and PL-enriched hepatocytes was performed in
NSG immunodeficient mice. Firstly, digitonin-collagenase enrichment of region-specific
hepatocytes was performed after 1 month of DEN-injection. After one month of
DEN-injection, liver tissues showed significant hyperpolyploidization but no preneoplastic
foci were detected, so it is a good stage for hyperpolyploid hepatocytes isolation without
preneoplastic foci contamination. 21 days after culture, CL- and PL-enriched hepatocyte
spheres were collected and implanted subcutaneously into the right and left flanks of each
mouse, respectively. As the results in **Fig. 3j-k** and **Supplementary Fig. 3g** show, right
flanks (CL-enriched hepatocytes) displayed significant clump formation 30 days
post-implantation, while no clumps were detected in the left flanks (PL-enriched
hepatocytes). The size of the CL-derived tumors (right) progressively increased to
become significantly larger than their PL counterparts (left) 50 days after injection (**Fig. 3j**).
Histological analysis with H&E staining revealed severe angiogenesis within the tumors
(**Fig. 3k**), and immunohistochemistry also showed high expression of the cell proliferation
marker Ki-67 within the tumor cells (**Supplementary Fig. 3g**). While a very low number of

small-size hepatocytes were observed in the CL-fraction during isolation, but CL-enriched
hepatocytes indeed display higher potential to generate spheres and tumors than
PL-enriched hepatocytes, suggesting the critical role of hyperpolyploid hepatocytes in
tumorigenesis. This new *in vivo* study allowed us to demonstrate that CL hepatocytes
have higher tumorigenicity than PL hepatocytes after DEN-treatment. Together, these
results support our hypothesis that CL-derived hyperpolyploid hepatocytes play an
important role in generation of preneoplastic cells after DEN treatment.

To address the second part of this reviewer's comment, we also isolated PL- and
CL-enriched hepatocytes from the liver without DEN treatment. Surprisingly, the new **Fig.**
**3g** shows that both PL- and CL-derived hepatocytes without DEN treatment have the
ability to form colonies. However, these colonies were seldom observed in untreated
hepatocytes compared to DEN-treated liver. Nevertheless, PL-enriched hepatocytes were
more likely to generate colonies than CL-enriched hepatocytes under normal conditions.
Similar findings have been reported by Katsuda *et. al.* this year and demonstrated that 2c
and 4c hepatocytes have higher colony formation ability *in vitro* [7], which is consistent
with the lower DNA content property of PL-enriched hepatocytes.

We added one paragraph describing the above findings in the manuscript,
accompanying new **Fig. 3j-k** and **Supplementary Fig. 3g**, as follows.

(main text)

"To address the tumorous potency of CL and PL hepatocytes after DEN treatment *in*
*vivo*, subcutaneous implantation of spheres derived from CL- and PL-enriched
hepatocytes into the right and left flanks of NSGTM immunodeficient mice was performed,
respectively. Clumps were found in the right flanks (CL-enriched hepatocytes) 30 days
post-injection, became significantly larger than those observed in the left flanks at 50 days
of injection, and further grew thereafter, while no clumps were detected in the left flanks
(Fig. 3j). H&E staining and immunohistochemistry revealed severe angiogenesis and high
expression of the cell proliferation marker Ki-67 within the clumps, confirming the
tumorigenicity of CL hepatocyte-derived spheres after DEN treatment (Fig. 3k and
Supplementary Fig. 3g)." (line 265-275)

3. Fig. 5d-f and associated supporting figure. It is interesting that DEN treatment leads to the formation of micronuclei and nuclear budding and that these effects are ameliorated by AZD. However, it is a conceptual leap to imply this is a mechanism of genome reduction. Consider the following examples:

** Page 17 (line 332): "Together with our finding presented in Fig. 3, these results paint a picture in which aberrant AURKB-induced hyperploidy plays a role in the transformation of CL-hepatocytes into preneoplastic cells via genome reduction."*

** Page 22 (line 427): "According to colony formation assay, our*

*findings support the hypothesis that CL-hepatocytes may be the*
*origin of HCC cells through genomic content reduction²².”*
** Fig. 8. The part about “genomic content reduction” is misleading.*
*There is no evidence that genome reduction is happening or that*
*any particular process is driving genome reduction. Direct evidence*
*must be presented, or the entire model needs to be revised. It would be*
*a lot of work, but it might be possible to demonstrate ploidy*
*reduction using the elegant lineage tracing model that was recently*
*described by Matsumoto and Grompe (Cell Stem Cell 2020;26:34).*

**Reply:**

We agree with the reviewer that Grompe’s group set up a great system for detecting
genomic content change, and the Rosa-Confetti mice would be a perfect model to address
the genome reduction issue. However, the current pandemic and time limitation made it
difficult for us to acquire the Rosa-Confetti mice for lineage tracing experiment, so we
attempted to address this comment by subcutaneous implantation of DEN-treated
zone-specific hepatocytes. One month after DEN injection, mice were sacrificed for
zone-specific hepatocytes isolation followed by sphere formation for 21 days. Spheres
derived from ~2000 CL or PL hepatocytes were then used for subcutaneous implantation
in 100 µl 50% Matrigel/media without serum, as described in reply #1. As shown in Fig.
3k and l. Three months after implantation, tumor nodes derived from CL hepatocytes were
examined. The tumor cells originating from hyperpolyloid CL hepatocytes, had
significantly smaller nuclei compared to those of hepatocytes and preneoplastic foci cells
in DEN treated liver (**Fig. 3l**). Interestingly, multipolar dividing cells reported by Grompe’s
group as the reversion mechanism of polyploid hepatocytes were detected (around 5%
incidence rate) within CL hepatocyte-derived tumors (higher magnification of **Fig. 3k**) [8-9].
This raises the possibility that “reductive mitoses” can happen in CL hepatocyte-derived
tumors and results in polyploidy reversal to produce daughter cells with lower DNA content.
By combining the evidences shown in **Fig. 3k-l**, we propose that CL hyperpolyloid
hepatocytes are the likely origin of the small preneoplastic cells in DEN-treated liver.
Because direct evidence for a genome reduction event is still lacking, we toned down the
description in Page 17 (line 332), Page 22 (line 427), and the section about “genomic

content reduction” in Fig. 8.

We added one paragraph describing the above findings in the manuscript,
accompanying new **Fig. 3k and I**, as follows.

(main text)

“To investigate whether hyperpolyploid hepatocytes are the origin of the smaller
preneoplastic cells, the size of the nuclei of cells in the clump was further analyzed. During
isolation of CL hepatocytes, very few cells with small nuclei were observed, but 80 days
after implantation cells in the CL hepatocytes-derived clumps had significantly smaller
nuclei compared to CL hepatocytes in DEN-treated liver (Fig. 3I). Interestingly, multipolar
dividing cells reported by Grompe’s group as the reversion mechanism of polyploid
hepatocytes were detected (around 5% incidence rate) within CL hepatocytes-derived
tumors (higher magnification images of Fig. 3k)^{36,37}. This observation raises the possibility
that “reductive mitoses” can happen in CL hepatocyte-derived tumors and results in
polyploidy reversal to produce daughter cells with lower DNA content. Together these
results show that hyperpolyploid hepatocytes have high tumorigenicity and may be the
origin of the small preneoplastic cells seen after DEN treatment.” (line 275-287)

“Together with our findings presented in Fig. 3, these results paint a picture in which
aberrant AURKB-induced hyperpolyploidy also plays a role in the transformation of CL
hepatocytes into preneoplastic cells via subsequent reduction of nucleus size, the sign of
a probable genome reduction event.” (line 377-380)

“According to *in vitro* and *in vivo* assays presented in Fig. 3g-l, our findings support
the hypothesis that CL hepatocytes, after a reductive mitosis or other uncharacterized
genome reduction events, may be the origin of HCC cells.” (line 468-470)

4. It is shown in Fig. 2 that hepatocytes with vacuoles are enriched in the CL region, and this is interpreted as an indicator of senescence and precancerous phenotype. This is fairly basic characterization. What else is happening to the cells? For example, do they have other features of senescent cells (e.g., beta-gal expression, shortened telomeres); do they have damaged DNA or oxidative damage? A more detailed characterization is required.

Reply:

To address this comment, we have checked a DNA damage marker, γ H2AX, and a senescence marker, Lamin B1 in control and DEN-treated liver. γ H2AX has been reported as a sensitive molecular marker of DNA damage and repair[10], and loss of Lamin B1 is a senescence-associated biomarker linked to certain aspects of tumor progression through large-scale changes in gene expression [11-12]. As shown in **Supplementary Fig. 2d-f**, γ H2AX expression was significantly upregulated in DEN-treated liver, comparing to control group. Importantly, in DEN-treated liver, higher expression of γ H2AX was detected specifically around CL hepatocytes and preneoplastic foci, indicating the severe DNA damage in these regions after DEN treatment. Additionally, senescence was assessed by the expression level of Lamin B1. DEN treatment caused downregulation of Lamin B1 in hepatocytes, but not in portal triad endothelial cells (yellow arrow). Importantly, lowest level of Lamin B1 expression were seen in preneoplastic cells, compared to that in adjacent hepatocytes.

We added one paragraph describing the above findings in the manuscript, accompanying new **Supplementary Fig. 2d-f**, as follows.

*(main text)*

“Furthermore, senescence and DNA damage were examined by Lamin B1 and
γ H2AX, respectively (Supplementary Fig. 2d-f). In DEN-treated liver, higher expression of
γ H2AX was detected specifically around CL hepatocytes and preneoplastic foci, indicating
severe DNA damage in these regions (Supplementary Fig. 2d,f). In contrast with γ H2AX,
downregulation of Lamin B1 was observed in DEN-treated liver, which was specifically
identified in hepatocytes but not in other cell types such as portal triad endothelial cells
(yellow arrow). Importantly, lowest Lamin B1 expression was seen in preneoplastic cells,
compared to adjacent hepatocytes (Supplementary Fig. 2e,f)” (line 211-219)

**Minor concerns from reviewer #1**

*1. Work from Bou-Nader, et al. (Ref 21) is mentioned in the*
*Introduction (page 6; line 111). The paper shows that human HCC*
*are characterized by increased nuclear ploidy. This is an*
*appropriate paper to cite, but it is important to emphasize that the*
*tumors with increased ploidy have mutations in p53. This*
*should be added to the text.*

**Reply:**

We have modified the corresponding sentence as follow.

(Main Text)

“Recent study revealed that hyperploidy hepatocytes are associated with worse
prognosis in human liver HCC, and that HCCs characterized by a low degree of
differentiation and TP53 mutations have higher levels of ploidy²¹.” (line 111-114)

*2. Page 6 (lines 114-116). What type(s) of “hyperploidy giant*
*cell” cells are being described here? Without mentioning the*
*cell or tissue type, the next sentence doesn’t make sense.*

**Reply:**

We now specify these cells as follow.

(Main Text)

“Hyperploidy giant cells, such as human ovarian, breast, colon, and prostate
cancer cell lines, have been demonstrated to serve as a source of stemness and tumor
heterogeneity through genomic reduction pathways²⁵.” (line 116-118)

*3. The designation of ploidy types in Supplementary Fig 1b is not*
*accurate. The math at the bottom makes sense assuming the cells*
*are in the same phase of the cell cycle (e.g., in G0/G1). However, if*
*some cells have replicated their DNA and some have not, then the*
*scaling does not make sense. I suggest using “chromatid number”*
*(c) instead of “chromosome number” (n). For instance, in the case*
*of a cell with tetraploid content, it could be either a tetraploid cell in*

G0/G1 (i.e., 4n4c) or a diploid in G2/M (i.e., 2n4c). To avoid confusion, it is safest to call the population 4c. I suggest labeling the populations in Supporting Figure 1b as 2c, 4c, 8c, etc. Ploidy populations in the FACS plots should be renamed as well.

**Reply:**

Thanks for reviewer's suggestion, we agree with this opinion and have change chromosome number (n) to chromatid number (c) in **Supplementary Fig. 1b** and FACS plots in Fig. 3f and **Supplementary Fig. 3a**

b

f

4. Live cell imaging is used to illustrate patterns of cell division by hepatocytes from control and DEN-treated mice. One can generally appreciate the patterns of cytokinesis described in the text, but it is very difficult to see cell boundaries and to appreciate the outcome of each division. It is especially difficult to appreciate abscission failure. Technical limitations of imaging include the following: (a) use of phase contrast only (can markers be used to highlight microtubules or cell membranes?); (b) cells are packed densely; (c) duration of imaging is limited. For example, in Supplementary Video

*4, one can see the first-round abscission failure, but the*
*second-round failure is not convincing because the movie ends so*
*quickly. In Supplementary Video 5, it is hard to appreciate that*
*cytokinesis is complete. I fully recognize that live cell imaging of*
*primary hepatocytes is challenging, but the data would be much*
*more convincing with clearer, less ambiguous movies. Additionally,*
*it would be helpful to add time stamps to the movies.*

**Reply:**

Actually, we have attempted to use GFP and RFP to outline cell morphology and
detail structure of cytokinesis, but phototoxicity caused hepatocytes to die gradually
during long-term live imaging. We then used phase contrast under 20x magnification for
high resolution image recording. To prevent ambiguous images caused by high cell
density, we have also observed lower culture density with 2×10^5 cells per well of a 6-well
plate. However, proliferating cells eventually led to increased of cell density during
long-term recording, which is the main cause of ambiguous images at the late recording
stage. We hope the reviewer will find the new **Supplementary Video 4** and **5**, showing
different cells and with added time stamps, to be more convincing.

*5. Tumor nodules at 1, 2 and 3 months post DEN injection are*
*continuously classified as “preneoplastic” lesions. What criteria are*
*the authors using to distinguish between “preneoplastic” and*
*cancerous lesions. For example, is the small nodule in Fig. 1b*
*preneoplastic or is it a tumor? The authors should describe their*
*classification system.*

**Reply:**

The development of primary tumors is often preceded by a sequence of histogenic
and molecular alternations in humans and experimental rodents. The sequentially
pathological changes have been well described in human and rodents, and the
terminology and criteria for various lesions has been defined based on consensus
diagnosis [13]. The first visible pathological lesion in the progression of cancer can be
classified as a preneoplastic lesion, characterized by as a small focus of cellular

alternation and appear to represent clonal expansion of about a hundred cells. These
preneoplastic lesions are difficult to detect with the naked eyes but can be identified
microscopically. However, tumor nodules, constituted by continuously growing of
preneoplastic cells, can be observed directly through naked eyes on the tissue surface.
Tumor nodule may identify as benign tumor since they grow by expansion of
preneoplastic cells and are not invasive with well differentiated morphology. In **Figure 1b**,
the whitish and round clump on the liver surface can be defined as a tumor nodule.

We added a new paragraph describing in the Supplementary material as follows.

*(Supplementary material)*

“Definition of preneoplastic lesion

The sequentially pathological changes have been well described in human and
rodents, and the terminology and criteria for various lesions has been defined based on
consensus diagnosis. The first visible pathological lesion in the progression of cancer can
be classified as a preneoplastic lesion, characterized by as a small focus of cellular
alternation and appear to represent clonal expansion of about a hundred cells. These
preneoplastic lesions are difficult to detect with the naked eyes but can be identified
microscopically. For tumor nodules, constituted by continuously growing of preneoplastic
cells, can be observed directly through naked eyes on the tissue surface. Tumor nodule
may identify as benign tumor since they grow by expansion of preneoplastic cells and are
not invasive with well differentiated morphology. For instance, in Figure 1b, the whitish
and round clump on the liver surface can be defined as a tumor nodule.” (line 200-211)

*6. The writing is overall very good, but there are a number instances*
*throughout the text where an incorrect word is used. These issues*
*should be addressed.*

**Reply:**

Thank you very much for appreciation comment and the support to our writing. As the
reviewer pointed out the incorrect word using, we have checked and corrected these
errors and written in blue.

**For reviewer #2:**

We would like to thank you for the time and valuable remarks. As described hereafter,
we have invested great efforts in order to improve the manuscript in light your comments.
Several major additions have been made point-by-point as follows.

**Major comments**

Central limitations of this study are:

1. The authors show an altered behaviour of polyploid hepatocytes after DEN, and reduced 'preneoplastic;' burden after AZD1152. We do not know however which of the many abnormal nodules which form in DEN go onto tumours and how representative they are of human disease progression, especially in the face of cirrhosis. The authors do not show that either the polyploid hepatocytes or the smaller preneoplastic foci are genuine precursors of cancer in this model (transplant experiments could demonstrate this), or that end stage tumour/disease burden is altered upon early AZD1152 treatment. Showing either of both of this would significantly increase the impact of the paper.

**Reply:**

We have performed transplantation experiments as suggested. To address the tumorous potency of CL and PL hepatocytes *in vivo*, subcutaneous implantation of spheres derived from CL- and PL-enriched hepatocytes was performed in NSG immunodeficient mice. Firstly, digitonin-collagenase enrichment of region-specific hepatocytes was performed after 1 month of DEN-injection. After one month of DEN-injection, liver tissues showed significant hyperpolyploidization but no preneoplastic foci were detected, so it is a good stage for hyperpolyploid hepatocytes isolation without preneoplastic foci contamination. 21 days after culture, CL- and PL-enriched hepatocyte spheres were collected and implanted subcutaneously into the right and left flanks of each mouse, respectively. As the results in **Fig. 3j-k and Supplementary Fig. 3g** show, right flanks (CL-enriched hepatocytes) displayed significant clump formation 30 days post-implantation, while no clumps were detected in the left flanks (PL-enriched hepatocytes). The size of the CL-derived tumors (right) progressively increased to

390 become significantly larger than their PL counterparts (left) 50 days after injection (**Fig. 3j**).
Histological analysis with H&E staining revealed severe angiogenesis within the tumors
(**Fig. 3k**), and immunohistochemistry also showed high expression of the cell proliferation
marker Ki-67 within the tumor cells (**Supplementary Fig. 3g**). While a very low number of
small-size hepatocytes were observed in the CL-fraction during isolation, but CL-enriched
hepatocytes indeed display higher potential to generate spheres and tumors than
PL-enriched hepatocytes, suggesting the critical role of hyperpolyploid hepatocytes in
tumorigenesis. This new *in vivo* study allowed us to demonstrate that CL hepatocytes
have higher tumorigenicity than PL hepatocytes after DEN-treatment. Together, these
results support our hypothesis that CL-derived hyperpolyploid hepatocytes play an
important role in generation of preneoplastic cells after DEN treatment.

Additionally, the tumor cells originating from hyperpolyploid CL hepatocytes, had
significantly smaller nuclei compared to those of hepatocytes and preneoplastic foci cells
in DEN treated liver (**Fig. 3l**). Interestingly, multipolar dividing cells reported by Grompe's
group as the reversion mechanism of polyploid hepatocytes were detected (around 5%
incidence rate) within CL hepatocyte-derived tumors (higher magnification of **Fig. 3k**) [8-9].
This raises the possibility that "reductive mitoses" can happen in CL hepatocyte-derived
tumors and results in polyploidy reversal to produce daughter cells with lower DNA content.
By combining the evidences showed in **Fig. 3k-l**, we propose that CL hyperpolyploid
hepatocytes are the likely origin of the small preneoplastic cells in DEN-treated liver.
Because direct evidence for a genome reduction event is still lacking, we toned down the
description in line 375 and 468, and the section about "genomic content reduction" in Fig.
8.

To address whether early AZD1152 treatment ameliorates DEN-induced
preneoplastic formation, AZD1152 (50mg/kg body weight/ 2 days) was injected into
DEN-treated mice intraperitoneally one week after DEN injection, and the results have
been presented in **Fig. 7** and **Supplementary Fig. 7** previously. We addressed the effect
of AZD1152 on the numbers of BrdU positive signal, vacuolation, and preneoplastic foci
and found that ADZ1152 treatment indeed significantly reduces DEN-induced pathology in
the liver. As shown in new **Supplementary Fig. 7f**, the end stage of tumor burden was
investigated after 6 months of DEN-injection. DEN-caused enlarged tumor nodules were

observed on the surface of DEN-treated liver. Although DEN-injection accompanied two
422 months of early AZD1152 treatment also displayed tumor nodules on the surface of liver,
the size and the numbers of tumor nodule were extremely reduced. This indicates that
early AZD1152 treatment indeed ameliorates DEN-induced tumor burden. In sum,
AZD1152 injection not only significantly inhibits early preneoplastic lesions, but also
relieves end stage tumor burden in DEN-treated liver.

We added paragraphs describing the above findings in the manuscript,
accompanying new **Fig. 3j-l**, **Supplementary Fig. 3g**, and **Fig. 7f**, as follows.

*(main text)*

“To address the tumorous potency of CL and PL hepatocytes after DEN treatment in
vivo, subcutaneous implantation of spheres derived from CL- and PL-enriched
hepatocytes into the right and left flanks of NSGTM immunodeficient mice was performed,
respectively. Clumps were found in the right flanks (CL-enriched hepatocytes) 30 days
post-injection, became significantly larger than those observed in the left flanks at 50 days
of injection, and further grew thereafter, while no clumps were detected in the left flanks
(Fig. 3j). H&E staining and immunohistochemistry revealed the severe angiogenesis and
high expression of the cell proliferation marker Ki-67 within the clumps, confirming the
tumorigenicity of CL hepatocyte-derived spheres after DEN treatment (Fig. 3k and
Supplementary Fig. 3g). To investigate whether hyperpolyploid hepatocytes are the origin
of the smaller preneoplastic cells, the size of the nuclei of cells in the clump was further
analyzed. During isolation of CL hepatocytes, very few cells with small nuclei were
observed, but 80 days after implantation cells in the CL hepatocytes-derived clumps had
significantly smaller nuclei compared to CL hepatocytes in DEN-treated liver (Fig. 3l).
Interestingly, multipolar dividing cells reported by Grompe’s group as the reversion
mechanism of polyploid hepatocytes were detected (around 5% incidence rate) within CL
hepatocytes-derived tumors (higher magnification images of Fig. 3k)^{36,37}. This observation
raises the possibility that “reductive mitoses” can happen in CL hepatocyte-derived tumors
and results in polyploidy reversal to produce daughter cells with lower DNA content.
Together these results show that hyperpolyploid hepatocytes have high tumorigenicity
and may be the origin of the small preneoplastic cells seen after DEN treatment.” (line
265-287)

“Together with our findings presented in Fig. 3, these results paint a picture in which
 aberrant AURKB-induced hyperploidy also plays a role in the transformation of CL
 hepatocytes into preneoplastic cells via subsequent reduction of nucleus size, the sign of
 a probable genome reduction event.” (line 377-380)

“Moreover, AZD1152 treatment also partially reduced the size of the larger tumor
 nodules observed on the surface of the liver 6 months after DEN treatment
 (Supplementary Fig. 7f)”. (line 420-422)

“According to *in vitro* and *in vivo* assays presented in Fig. 3g-l, our findings support
 the hypothesis that CL hepatocytes, after a reductive mitosis or other uncharacterized
 genome reduction events, may be the origin of HCC cells.” (line 468-470)

2. *The relevance of the DEN mouse model to human HCC. This could be mitigated by alternative HCC models in the mouse. This discussion of accuracy of animal models is much debated in the field currently and there is no simple solution experimentally currently. I would recommend this point is discussed by the reviewers in greater detail but I do not see that repetition of this work in additional model systems if required for this current body of work. The major changes of DEN resulting in nuclear enlargement and polyploidy in the centrilobular area may well be a reflection of this being the area of principle cellular damaged after acute DEN administration. This should be discussed, but the data on reversal of ploidy and reduction of ‘preneoplastic foci’ with AURKB inhibition reassures that at least ploidy is related either directly or indirectly to foci formation.*

Reply:

This comment was very similar to the first comment of the first reviewer, who called
 for alternative HCC models. In response to these comments, three different mouse
 models were applied to address this question, including aflatoxin B1 (AFTB1) model [1],
 CCl₄ model [2], and high-fat diet (HFD) model [3]. These three model systems were

selected for the following reasons. Firstly, AFTB1 is a common contaminant produced by
*Aspergillus flavus* and *A. parasiticus* in a variety of foods grains. It has been demonstrated
as a very potent carcinogen causing HCC in humans and rodents [4]. Secondly, CCl₄ one
of the most potent hepatotoxins causing HCC, was widely used as a cleaning fluid in the
textile industry, and in households as a spot remover for fabrics, which is another most
potent hepatotoxins caused HCC [5]. Finally, high-calorie diet caused nonalcoholic fatty
liver disease (NAFLD) is a widespread metabolic disorder that is reported as a risk factor
for HCC [6]. Additionally, high-fat diet induced hyperpolyploid hepatocytes via oxidative
stress in mouse has been demonstrated by Desdouets's group in 2015, but they did not
specify the distribution of these cells [3]. The results obtained with these three mouse
models are now showed in **Supplementary Fig. 1k**. We found that DEN-induced
hyperpolyploid CL hepatocytes obtained with DEN can also be observed in AFTB1-, CCl₄-,
and HFD-treated liver. As shown in **Supplementary Fig. 1k**, hepatocytes under these
three conditions displayed an enlarged nucleus size near the central vein region
compared to control group. These results indicate that hyperpolyploidization of
hepatocytes around the central vein region is not limited to DEN but appears to be a
common event elicited by known HCC-causing xenobiotics and oxidative stress such as
AFTB1, CCl₄ and HFD. Xenobiotics and oxidative stress being the critical contributors to
HCC, these new findings strengthen the relevance of our study.

We added one paragraph describing the above findings in the manuscript,
accompanying new **Supplementary Fig. 1k**, as follows.

*(main text)*

"To investigate whether hyperployploidization of CL hepatocytes is simply an artifact
of the DEN model or a relevant feature of early-stage HCC formation caused by
xenobiotics or oxidative stress, three different mouse models, using Aflatoxin B1 (AFTB1),
carbon tetrachloride (CCl₄), and 45 kcal% high-fat diet (HFD), were used to address this
question. As shown in Supplementary Fig. 1k, hepatocytes in AFTB1- and CCl₄-treated
mice displayed enlarged nucleus size near the central vein region, compared to control
group. After 90 days of HFD treatment, hepatocytes also exhibited slightly larger nucleus
size than age-matched control nearby central vein region, but not in the liver with 60 days
of HFD treatment. These results indicated that not only DEN, but also xenobiotics and

HFD-induced oxidative stress specifically target CL and ML hepatocytes and cause
hepatic hyperpolyploidization within these two regions. More generally, the metabolic
sensitivity of CL and ML hepatocytes to xenobiotics is likely an important factor in HCC
development.” (line 162-175)

*3. If polyploidy hepatocytes are the source of HCC then*
*mechanisms of genetic reduction remains unanswered. The*
*authors highlight this appropriately but further work in this areas*
*would be welcome in the future. Clearly a link between*
*polyploidisation in early cancer and genetic-reduction in cancer*
*formation would provide a significantly more exciting and complete*
*study.*

**Reply:**

We thank the reviewer for this comment. This is what we are interested and intend to
investigate in the near future.

**Additional points include:**

*1. The introduction discusses genetic lineage tracing studies from*
*hepatocytes – it should be highlighted that these are in mouse*
*models. The introduction should not leave the reader with the idea*
*that in human other sources of HCC have been excluded.*

**Reply:**

Thanks for reviewer's suggestions, we have emphasized that these lineage tracing
study are from mouse models as below.

(Main Text)

"Genetic lineage-labeled tracing approaches showed that HCC and hepatocellular
adenoma (HCA) are derived from mature hepatocytes **in mouse model**^{6,7}. Moreover, adult
hepatocytes can trans-differentiate into biliary-like cells during liver cancer formation, and
de-differentiate into progenitor-like cells in p53 deficient **mouse liver**^{8,9}. **The relevance of**
**these models for human HCC is still a matter of debate.**" (line 89-94)

*2. The timepoint of non-DEN treated control should be stated in*
*each figure. As polyploidy changes over time in the mouse liver it is*
*important to compare non-DEN and DEN at matched timepoints*
*throughout. From my understand the controls used are aged*
*matched to p15+3M but this should be explicitly stated and ideally*
*other aged match controls used when comparing the other*
*timepoints. Similarly non-DEN control should be clearly identifiable*
*in the key figure 5G.*

**Reply:**

We agree with reviewer's comment, and we have now added control groups back in

the **Supplementary Fig. 1c**, which match with the timepoints of DEN-treated groups. As
 shown in **Supplementary Fig. 1c**, the size of PL hepatocytes nuclei did not significantly
 change during liver development. Developmentally, however, the effect of age on the size
 of nuclei in CL hepatocytes became evident around p105 of age in control mice. The
 hepatocytes from older control mice at p105 had larger nuclei compared to younger
 control mice, but DEN-treated hepatocytes still display significantly bigger nuclei than
 age-matched control mice. Additionally, when the reviewer mentions Fig. 5g, does he/she
 mean Fig. 7, since there is no Fig. 5g in our manuscript? If it is Fig. 7g indeed, we have
 already included non-DEN control in our original text.

We added one paragraph describing the above findings in the manuscript,
 accompanying new **Supplementary Fig. 1k**, as follows.

(Main Text)

“The hepatocytes from older control mice at p105 had larger nuclei compared to
 younger control mice, but DEN-treated hepatocytes still display significantly bigger cells
 and nuclei than age-matched control mice (Fig. 1c,d and Supplementary Fig. 1b-g), in line
 with previous findings.” (line 145-148)

*3. Additional detail is required in the methods. In what vehicle was*
*the AZD1152 delivered? Which stock of BrdU was used?*

**Reply:**

For AZD1152, in the beginning, we followed previous report and tried that AZD1152
powder was dissolved in 3M Tris, pH 9.0 buffer to obtain a solution at the concentration of
2.5 mg/ml; however, it was very difficult to get clear AZD1152 solution. Then, we used
DMSO as the solvent to get the stock with concentration of 10 mg/ml. AZD1152 was
finally diluted into working concentration by normal saline for further study.

For BrdU, we purchased BrdU-labeling reagent from Life technologies (REF 000103/
LOT 1923353A) for ready to use. It shows concentration with 3 mg/ml on the company
website but not in the datasheet. The dosage we conducted was in mice injected
intraperitoneally with 10 µl of BrdU labeling reagent per gram of body weight according to
the user manual.

We have now added the necessary details in the **Supplementary Material** as follow.

*(Supplementary Material)*

“AZD1152 powder was dissolved in DMSO to get the stock with concentration of 10
596 mg/ml. Then, AZD1152 was finally diluted into working concentration by normal saline for
i.p. injection. For cell proliferation analysis *in vivo*, BrdU (10 µl/g body weight; 3 mg/ml)
purchased from company was directly used and injected i.p. for 5h.” (line 90-93)

*4. In general the figures are of good quality but the legends could*
*make it easier to understand what was done in the experiments and*
*what is being used as a control. Data in Figure 3G should be*
*compared to hepatocytes from non-DEN treated animals. Statistical*
*comparison of DEN and DEN & AZD1152 groups in Figure 5 should*
*be shown. Is significant reversal upon AZD treatment observed?*

**Reply:**

We apologize for not showing control experiment in **Fig. 3g**. Now, **Fig. 3g** shows the
potency of spheres formation of PL- and CL-enriched hepatocytes from the liver without
DEN treatment. Surprisingly, without DEN treatment both PL and CL hepatocytes have the
ability to form colonies. However, these colonies were seldom observed in untreated

hepatocytes compared to DEN-treated liver. Nonetheless, PL-enriched hepatocytes were
more likely to generate colonies than CL-enriched hepatocytes under normal conditions.
Similar findings have been reported by Katsuda *et. al.* this year and demonstrated that 2c
and 4c hepatocytes have higher colony formation ability *in vitro* [7], which is consistent
with the lower DNA content property of PL-enriched hepatocytes.

For the statistical comparison of DEN and DEN + AZD1152 groups in **Fig. 5e**, no
significant difference was observed in our original figure, but it is very close to $P = 0.05$
(Complete cytokinesis/ $P = 0.0561$; Abscission failure/ $P = 0.0630$). We added this
information to Fig. 5e as reviewer's request.

We added one paragraph describing the above findings in the manuscript (line
247-252), accompanying new **Fig. 3g**, as follows. For **Fig. 5e**, statistical meaning between
DEN and DEN + AZD1152 groups was added.

(main text)

“To examine whether CL hepatocytes displayed cancerous properties after DEN
injection, colony formation assay was performed with CL or PL hepatocytes, the numbers
of colonies analyzed on the 21st day *in vitro* (DIV). CL hepatocytes showed higher colony
formation ability compared to PL hepatocytes and to control hepatocytes isolated from the
liver without DEN treatment (Fig. 3g, Supplementary Fig. 3c, and Supplementary Video 6,
7).” (line 250-255)

**Stylistic points:**

1. The authors switch between days and months post DEN, consistency here would help the reader.

**Reply:**

We have changed all the labeling from W (weeks) and M (months) to p (postnatal
638 days) for the consistent presentation. Please check **Fig. 1, 2, 7 and Supplementary Fig 1,**
**2, 7.**

*2. Please tone down strongly categorical descriptions of the data*
*e.g. 'clearly indicates' change to 'indicates'.*

**Reply:**

As suggested we have toned down categorical descriptions in the manuscript.

(main text)

“This result ~~clearly~~ indicated that not only DEN, but also xenobiotics and HFD-induced
oxidative stress specifically target CL and ML hepatocytes and cause hepatic
hyperpolyploidization within these two regions.” (line 171-173)

“Altogether, our observations ~~clearly~~ demonstrate that aberrant activation of AURKB is a
critical pathway leading to DEN-induced hepatocyte hyperpolyploidization,...” (line
385-387)

*3. Similarly the complete lack of periportal polyploidization should*
*not be excluded unless this is convincingly shown (lines 218 and*
*359).*

**Reply:**

We appreciate the reviewer’s suggestion and agree with it. Our new *in vivo* data (Fig.
3j-m) indicated that CL hepatocytes express higher potential to generate small size tumor
cells, but it is still difficult to exclude the possibility of PL hepatocytes. In response to this
comment we have modified the main text as follows.

**(main text)**

“If this is the case, preneoplastic cells would appear **predominantly** among CL
hepatocytes.” (line 239-240)

“Like our previous findings (Fig. 1c,d and Supplementary Fig. 1c-e), DEN treatment
caused hyperpolyploidization of CL and ML hepatocytes **predominantly**.” (line 404-405)

*4. Additionally the comment ‘until now’ (line 416) referring to link to*
*HCC should be nuanced – please place into context of my major*
*comments 1 and 2. This also applies to the following sentence in*
*line 418.*

**Reply:**

We thank for reviewer’s suggestion, we have nuanced the corresponding section as
follows..

(main text)

“Moreover, mouse models have established a relationship between stresses and
pathological hyperpolyploidy in the liver¹⁷⁻²⁰, but the direct evidence of tumorigenesis are
lacking, ~~until now~~. Our results reveal ~~an intimate~~ a molecular correlation between
pathological hyperpolyploidy of CL hepatocytes and ~~xenobiotics-induced~~ HCC formation in
~~mouse~~.” (line 455-459)

*5. Labelling of tumour and non tumour is misleading e.g. Figure 2h.*

**Reply:**

We re-labeled tumor and non-tumor to Foci and Non-foci in Fig. 2h.

*6. It would be helpful to show increased DNA content in the enlarge*
*hepatocytes after DEN earlier and ideally related to the other data in*
*Figure 1.*

**Reply:**

As the reviewer’s mention DNA content need to be investigated, therefore, flow
cytometry was performed to examine DNA copy numbers of control and DEN-treated

livers at different time points. We found that the increased nucleus size was accompanied
by increased ploidy levels developmentally in both control and DEN-treated groups.
Importantly, in DEN-treated liver, the populations of hepatocytes with DNA content > 8n
were markedly accelerated, comparing to control group. Notable, population of 2n
hepatocytes was slightly increased in the liver after 3 month of DEN treatment, which may
be contributed by preneoplastic cells. These observations demonstrate that DEN-caused
enlarged nucleus size of hepatocytes and increase of DNA content is etiologically
connected.

To address this comment, we have added one paragraph describing the above
findings in the manuscript, accompanying new **Supplementary Fig 1d** showing new flow
cytometry data, as follows.

(main text)

“The hepatocytes from older control mice at p105 had larger nuclei compared to
younger control mice, but DEN-treated hepatocytes still display significantly bigger cells
and nuclei than age-matched control mice (Fig. 1c,d and Supplementary Fig. 1b-g), in line
with previous findings¹⁷. Flow cytometry analysis also demonstrated that enlarged nuclei
correlated with higher DNA content in DEN-treated hepatocytes (Supplementary Fig. 1d).”

(line 145-150)

d

**Minor comments**

*1. Does dual inhibition of AURKA and AURKB have a similar effect;*
*using ZM447439 in vitro for example. The addition of another*
*pharmacological inhibitor would strengthen the study. This need not*
*be completed throughout the experiments, but in some critical short*
*term in vitro studies it would add further weight to the authors*
*argument.*

**Reply:**

We thank reviewer for this suggestion. According to the datasheet, ZM447439 has an
IC50 of 110 nM and 130 nM for AURKA and AURKB, respectively. Therefore, we used two
different dosage of ZM447439, 150 nM and 500 nM, in time lapse image assay. Around 50
~ 100 dividing cells were analyzed from an experiment in each group. We analyzed an
entire cycle of cell division, and found that abscission failure was lower in control (8.06%
incidence rate) than in DEN-treated liver (19.35% incidence rate). After AZD1152
treatment, DEN-caused abscission failure can be rescued partially (11.764% incidence
rate) like our previous finding. Interestingly, 150 nM ZM447439 showed the similar
influence like AZD1152 treatment that ameliorates DEN-induced abscission failure of
hepatocytes (12.67% incidence rate); however, abscission failure ratio was increase in
control group with 150 nM ZM447439 treatment (18.16% incidence rate). We also found
that DEN-induced abscission failure cannot to be reduced by high dosage (500 nM)
treatment of ZM447439 (21.21% incidence rate).

AURKA/B isoforms have been implicated in the regulation of cell cycle progression,
and are essential during mitosis. The functional overlapping between AURKA/B isoforms
has been reported, and loss of AURKB caused cellular deficiency can be compensated by
AURKA through INCENP-dependent manner [14]. Moreover, AURK affinities can be
swapped by mutation of an amino acid residue of AURKA or AURKB, indicating that their
substrate specificities are similar and may compensate for each other's loss. Critically,
existing evidences also support that the expression and activity of AURKA/B are tightly
controlled, either up- or down-regulation of AURKA/B will cause chromosomal instability,
abscission failure and polyploidy formation [15-19]. According to previous reports and our
result, we hypothesize that the expression and balance of AURKA/B isoforms are

controlled tightly within a special range for processing correct mitosis. In control
 hepatocytes, 150 nM ZM447439 treatment may severely reduce both AURKA/B activity
 and cause abscission failure. However, in DEN-treated hepatocytes, 150 nM ZM447439
 treatment may not only reduce the activity of hyper-activated AURKB, but also make a
 new balance between AURKA and AURKB, which is acceptable to ensure accurate
 transition of mitosis and prohibit abscission failure. As the concentration of ZM447439
 increased to 500 nM, we speculated that abscission failure may be caused by severe
 reduction of AURKA/B activity, but not by DEN-induced upregulation of AURKB.

This preliminary data is for reviewer's reference, we didn't include it in the manuscript.

 **2. In our experience quantification of hepatocytes based solely on**
 **nuclei shape after immunofluorescence is challenging, larger**
 **leukocytes especially lymphocytes may also have larger nuclei and**
 **may be misclassified. The addition of a bona fide hepatocyte**
 **marker would help. Similarly whilst it is standard to quantify across**
 **the lobule from CV to PV the complex 3D nature of tissue may**
 **mean that zones are mis-defined. Large scale analysis of tissue**
 **helps to reduce this experimental 'noise' but ideally markers of**
 **zones could be used to define distinct areas rather than 2D**
 **morphology. Neither of these issues are necessary to address for**
 **revision but are mentioned as important to the field.**

**Reply:**

We would like to thank the reviewer for this thoughtful comments and efforts towards
improving our study. We will keep these comments in mind.

*3. The zonal proliferation observed in Fig 1G is discrepant with that*
*observed by others, notably that of Tchorz group. This does not*
*require revision but can the authors speculate why they see CL*
*dominant proliferation in homeostasis compared to ML observed by*
*others.*

**Reply:**

This is a critical but still unclarified issue of liver homeostasis. Tchorz's group used
AXIN2 lineage tracing to address that the role of AXIN2⁺ pericentral hepatocytes and the
contribution of other liver zones during homeostasis and repair [20]. Although they found
that hepatocytes throughout the liver can upregulate AXIN2 and LGR5 and contribute to
liver regeneration, pericentral hepatocytes consistently express AXIN2⁺ signal under
physiological condition, which indicates the dominant proliferation ability of pericentral
hepatocytes. However, why pericentral hepatocytes show higher proliferation properties is
still unclear. According to previous studies, the liver is responsible for the
biotransformation of xenobiotics and endogenous metabolic byproducts of metabolism
[21-23]. Histologically, the liver acts in a hierarchical fashion with the gradients of nutrients,
hormones, and oxygen. Since oxygen triggers the formation of reactive oxygen species
(ROS), an intra-acinar redox gradient exists causing higher levels of hypoxia-inducible
factors (HIFs) in the pericentral region. In this light, higher level of ROS caused oxidative
stress within pericentral region may increase the risk of hepatic damage and cell death in
this area. Keeping superior proliferative ability of pericentral hepatocytes over other
hepatocytes may be necessary to provide slowly cycling/renew and maintain the
parenchyma pool and ensure proper metabolic functions.

*4. Line 409 – is this Figure 8?*

**Reply:**

Yes, it is Fig. 8, we have corrected this typo.

5. Line 414 – the link to HCC disease progression is unclear from this sentence. Can this be amended for the clarity of the reader.

**Reply:**

We have modified sentence as follow.

*(main text)*

“Deficiencies in circadian genes also ~~showed to~~ increased abscission
failure-mediated hyperpolyploid CL hepatocytes and liver tumorigenesis^{13,48}, suggesting a
possible link between the formation of hyperpolyploid hepatocytes and an early stage of
HCC progression.” (line 452-455)

*6. There are minor typographical errors in the main manuscript and
figure legends which should be corrected.*

**Reply:**

We thank reviewer pointed out the typographical errors in main text and figure
legends. We have corrected these typographical errors.

*7. Figure 3f – is this Hoeschst 33324?*

**Reply:**

We used DAPI for nuclear staining in Fig. 3f. The membrane permeability of DAPI is
somewhat less than Hoechst dyes and must be used at a higher concentration (usually 10
817 µg/ml) for live cell staining, but it can be used for fixed cell staining at 1 µg/ml that is similar
to Hoechst33342. In Fig. 3f, after enrichment, region-specific hepatocytes were fixed
immediately by 4% PFA/ PBS, and then nuclei were stained with 2 µg/ ml DAPI at 4°C
overnight for further analysis.

**References**

- 1. Chawanthayatham, S., et al., *Mutational spectra of aflatoxin B1 in vivo establish*
*biomarkers of exposure for human hepatocellular carcinoma*. Proc Natl Acad Sci U S A,
2017. **114**(15): p. E3101-E3109.
- 2. Chen, F., et al., *Broad Distribution of Hepatocyte Proliferation in Liver Homeostasis and*
*Regeneration*. Cell Stem Cell, 2020. **26**(1): p. 27-33 e4.
- 3. Gentric, G., et al., *Oxidative stress promotes pathologic polyploidization in nonalcoholic*
*fatty liver disease*. J Clin Invest, 2015. **125**(3): p. 981-92.
- 4. Kucukcakan, B. and Z. Hayrulai-Musliu, *Challenging Role of Dietary Aflatoxin B1 Exposure*
*and Hepatitis B Infection on Risk of Hepatocellular Carcinoma*. Open Access Maced J Med
Sci, 2015. **3**(2): p. 363-9.
- 5. Masuda, Y., [*Learning toxicology from carbon tetrachloride-induced hepatotoxicity*].
Yakugaku Zasshi, 2006. **126**(10): p. 885-99.
- 6. Michelotti, G.A., M.V. Machado, and A.M. Diehl, *NAFLD, NASH and liver cancer*. Nat Rev
Gastroenterol Hepatol, 2013. **10**(11): p. 656-65.
- 7. Katsuda, T., et al., *Transcriptomic Dissection of Hepatocyte Heterogeneity: Linking Ploidy,*
*Zonation, and Stem/Progenitor Cell Characteristics*. Cell Mol Gastroenterol Hepatol, 2020.
**9**(1): p. 161-183.
- 8. Duncan, A.W., et al., *The ploidy conveyor of mature hepatocytes as a source of genetic*
*variation*. Nature, 2010. **467**(7316): p. 707-10.
- 9. Matsumoto, T., et al., *In Vivo Lineage Tracing of Polyploid Hepatocytes Reveals Extensive*
*Proliferation during Liver Regeneration*. Cell Stem Cell, 2020. **26**(1): p. 34-47 e3.
- 10. Mah, L.J., A. El-Osta, and T.C. Karagiannis, *gammaH2AX: a sensitive molecular marker of*
*DNA damage and repair*. Leukemia, 2010. **24**(4): p. 679-86.
- 11. Freund, A., et al., *Lamin B1 loss is a senescence-associated biomarker*. Mol Biol Cell,
2012. **23**(11): p. 2066-75.
- 12. Garvalov, B.K., S. Muhammad, and G. Dobrova, *Lamin B1 in cancer and aging*. Aging
(Albany NY), 2019. **11**(18): p. 7336-7338.
- 13. Henson, D.E. and J. Albores-Saavedra, *The pathology of incipient neoplasia*, in *Cancer*
*Chemoprevention*. 2001, Humana Press. p. 69-96.
- 14. Nguyen, A.L., et al., *Genetic Interactions between the Aurora Kinases Reveal New*
*Requirements for AURKB and AURKC during Oocyte Meiosis*. Curr Biol, 2018. **28**(21): p.
3458-3468 e5.
- 15. Sehdev, V., et al., *The combination of alisertib, an investigational Aurora kinase A inhibitor,*
*and docetaxel promotes cell death and reduces tumor growth in preclinical cell models of*
*upper gastrointestinal adenocarcinomas*. Cancer, 2013. **119**(4): p. 904-14.
- 16. Gomaa, A., et al., *Epigenetic regulation of AURKA by miR-4715-3p in upper*
*gastrointestinal cancers*. Sci Rep, 2019. **9**(1): p. 16970.

- 17. Walsby, E., et al., *Effects of the aurora kinase inhibitors AZD1152-HQPA and ZM447439 on*
*growth arrest and polyploidy in acute myeloid leukemia cell lines and primary blasts.*
*Haematologica*, 2008. **93**(5): p. 662-9.
- 18. Nguyen, H.G., et al., *Deregulated Aurora-B induced tetraploidy promotes tumorigenesis.*
*FASEB J*, 2009. **23**(8): p. 2741-8.
- 19. Ricke, R.M., K.B. Jeganathan, and J.M. van Deursen, *Bub1 overexpression induces*
*aneuploidy and tumor formation through Aurora B kinase hyperactivation.* *J Cell Biol*, 2011.
**193**(6): p. 1049-64.
- 20. Sun, T., et al., *AXIN2(+) Pericentral Hepatocytes Have Limited Contributions to Liver*
*Homeostasis and Regeneration.* *Cell Stem Cell*, 2020. **26**(1): p. 97-107 e6.
- 21. Kietzmann, T., *Metabolic zonation of the liver: The oxygen gradient revisited.* *Redox Biol*,
2017. **11**: p. 622-630.
- 22. Soto-Gutierrez, A., et al., *Pre-clinical and clinical investigations of metabolic zonation in*
*liver diseases: The potential of microphysiology systems.* *Exp Biol Med (Maywood)*, 2017.
**242**(16): p. 1605-1616.
- 23. Gebhardt, R. and M. Matz-Soja, *Liver zonation: Novel aspects of its regulation and its*
*impact on homeostasis.* *World J Gastroenterol*, 2014. **20**(26): p. 8491-504.

Table 1. List of revised Figures

Items	Event	Revised
1.	Groups title change	Change “Control” group title to “Ctrl” in all Figures; change “Non-tumor” and “Tumor” to “Non-foci” and “Foci”, respectively, in Fig. 2h and Fig. 3a.
2.	Time labeling change	According to reviewer #1’s suggestion, we changed time labeling from “W (week)” and “M (month)” to “p (postnatal day)” to make labeling more consistent in all Figures. For instance, “DEN+1M” and “DEN+2M” changed to “DEN/ p45” and “DEN/ p75”, respectively; “p15 + 30 day” changed to “p45”; “6w” and “10w” changed to “45” and “p75”, respectively.
3.	Change Fig. 3f and Supplementary Fig. 1b and 3a labeling.	According to reviewer #1’s suggestion, we changed instead of “chromosome number (n)” to “chromatid number (c)” to avoid confusion in different cell cycle states.
4.	Change the model description in Fig. 8	According to reviewer #1’s suggestion, we toned down the description in Fig. 8 model by change “Genomic content reduction” to “Nucleus size reduction”.
5.	Add new figures and make rearrangement in Supplementary Fig. 1	According to reviewers’ comments, we added new figures as Supplementary Fig. 1c, d, and k. The original Supplementary Fig. 1c, d, e, f, g, and h were changed to Supplementary Fig. 1e, f, g, h, l, and j.
6.	Add new figures and make rearrangement in Supplementary Fig. 2	According to reviewer #1’s comment, we added new figures as Supplementary Fig. 2d, e, and f. The original Supplementary Fig. 2d, and e were changed to Supplementary Fig. g, and h.
7.	Add new figures in Supplementary Fig. 3	According to reviewers’ comments, we added new figures as Supplementary Fig. 3g.

8.	Add new control group in Fig. 3g	According to reviewer #1's comment, we added new control group in Fig. 3g.
9.	Add new figures in Fig. 3	According to reviewers' comments, we added new figures as Fig. 3j, k, and l.
10.	Add new statistical meaning in Fig. 5e	According reviewer #2's comment, we added new statistical meaning in Fig. 5e.
11.	Add new figure in Supplementary Fig. 7f	According reviewer #2's comment, we added new figure in Supplementary Fig. 7f and g.
12.	Add new figure legends	We added new figure legends of Fig 3j-l, Fig 5a-b, Fig 6a, Fig 7a, Supplementary Fig. 1c, d, and k, Supplementary Fig. 2d-f, Supplementary Fig. 3g, Supplementary Fig. 7f and g.

Table 2. List of revised Video

Items	Event	Revised
1.	Add time stamps	According to reviewer #1's comment, we added time stamps in all Videos.
2.	Change videos with good image quality	According to reviewer #1's comment, we replaced Supplementary Videos 4 and 5 with good quality images.

**Table 3. List of the critical revise of manuscript**

[revised manuscript text omitted]

		progression.
16.	Add new words in DISCUSSION (line 468-470)	According to in vitro and in vivo assays presented in Fig. 3g-l, our findings support the hypothesis that CL hepatocytes, after a reductive mitosis or other uncharacterized genome reduction events, may be the origin of HCC cells ^{25, 36-37} .
17.	Add new words in DISCUSSION (line 486-490)	Although we did not formally investigate the mechanisms underlying genomic reduction, we observed multipolar dividing in CL hepatocytes-derived tumor cells (Fig. 3). Moreover, frequent nuclear budding and micronuclei in hyperpolyloid hepatocytes were identified in vitro (Fig. 6),...
18.	Add new words in ACKNOWLEDGMENTS (line 523-529)	The authors are indebted to Dr. Yi-Hsuan Huang and Dr. Ruey-Bing Yang for supplying facilities and drugs for in vivo and in vitro studies. The authors thank the pathology core of Institute of Biomedical Sciences (IBMS) for paraffin blocks and H&E stain processing. The authors also thank the assistance from Ms. Tzu-Wen Tai and Ms. Chia-Chen Tai in the flow cytometry core facility of IBMS (AS-CFII108-113), Academia Sinica for analysis of flow cytometry. The authors are grateful to Mr. Kuan-Yu Chou and Ms. Show-Rong Ma in microscopes facility of IBMS for images acquisition.

REVIEWER COMMENTS

Reviewer #1 (Remarks to the Author):

The authors performed numerous experiments to address concerns raised by both reviewers, and manuscript has been extensively revised. Most of my concerns have been addressed -- through a combination of new experimental data and text editing (where some of the sweeping conclusions were toned down). One remaining concern needs to be addressed. A major conclusion of the paper is summarized in the Introduction (lines 121-123):

"Hyperpolyploid CL hepatocytes are an oncogenerative source of preneoplastic cells through uncharacterized genome reduction processes."

I agree this is the mostly likely conclusion from the data. However, despite abundant supporting data, there is no direct evidence that hyperpolyploid CL hepatocytes are the source. I think it is important to revise the sentence in a manner similar to this: "CL hepatocytes, which are highly enriched with hyperpolyploid cells, are the source of preneoplastic cells, likely through uncharacterized genome reduction processes." The authors should also edit any other relevant sections/statements accordingly.

Reviewer #2 (Remarks to the Author):

I am grateful for the additional work provided by the authors in the revision of the manuscript. The additional transplantation experiments are particularly welcome and are a significant addition. Overall these revisions have improved the manuscript significantly.

In response to my comments.

1. The additional of the transplant experiment as suggested by myself and reviewer #1 is a helpful addition. This still fails to show direct evidence that the large cells themselves go on to form tumors in the transplant model as this is still a mixed population which is injected.

It is crucial that it is explicitly stated that the spheres injected in these assays are appropriately controlled. "Spheres derived from 2000 numbers of hepatocytes were used for implantation." This implies equal hepatocytes were plated and any resulting spheres were injected. There needs to be appropriate controls injecting equal numbers of spheres from each condition to show that some spheres are more aggressive depending on from where they were obtained.

As there are still smaller hepatocytes in the extracts from CL it is still possible that these are actual origin of the tumors. The evidence presented associatively links the two and the demonstration of frequent multipolar mitosis is supportive, however the possibility still exist that the smaller not the larger cells are those which give rise to foci. Whether these smaller cells' predecessors were ever larger remains to be seen. This could be discussed more clearly and the current wording revised: "Although we did not formally investigate the mechanisms underlying genomic reduction": Neither the mechanisms nor an absolute link were proven in this report.

"Together these results show that hyperpolyploid hepatocytes have high tumorigenicity and may be the origin of the small preneoplastic cells seen after DEN treatment." There was still no direct link between the hyperpolyploid hepatocytes themselves and the foci.

"In conclusion, here we revealed that, after DEN-induced carcinogenic liver injury, hyperpolyploid hepatocytes arising from abscission failure are oncogenerative cells and a major source of preneoplastic lesions." should be revised to "In conclusion, our data strongly suggests that, after DEN-induced carcinogenic liver injury, hyperpolyploid hepatocytes arising from abscission failure are oncogenerative cells and a major source of preneoplastic lesions."

Consistency re AZD injection. The manuscript states 1 week after DEN but the methods state 1 day after DEN.

Similarly the long term effects of early AZD1152 treatment on carcinogenesis are welcome and a helpful addition to the revised manuscript. I would recommend this important data on the prevention of the cancer phenotype is incorporated in the main Figures.

2. The additional models, again suggested by Reviewer #1 also, are helpful. However, CCl4 is an agent which like DEN causes pericentral injury and DNA damage. NAFLD is a disease which again is predominantly pericentral in fat loading. Aflatoxin B1 additionally requires activation by specific CYPs which are also zoned. Hence in the models shown it is unclear whether the hyperpolyploidisation is a feature of injury of early carcinogenesis. This should be discussed

This new data Fig S1k should be quantified.

3. I accept the author's comments that this is outwith the scope of the current manuscript.

The additional details are welcome to address the DEN timepoint series and the methodology.

4. (Additional points) The lack of significance appears to be a type 2 error due to underpowering of the experiments. Now that it is apparent that neither of these key results reach a reasonable cut off for significance I would suggest increasing the replicates to test whether there is a genuine and demonstrable effect on complete cytokinesis and abscission failure.

I agree with the changes made in response to my minor concerns.

Further proof reading and corrections are required e.g. Line 176 "The understand the cause of", and the supplemental information

I agree with the comments from reviewer #1, which are broadly in line with my own. I would, in response to the authors reply, highlight that the additional markers of laminB1 and gh2Ax are helpful. However, I would caution the use of "senescence" without direct evidence of lack of proliferation in these cells and encourage the authors not to imply that cells in this area, which is the most proliferative are senescence. LaminB1 is associated with senescence but is not a bona fide marker of senescence – few are, as the authors will be aware.

The delineation of preneoplastic and neoplastic is still arbitrary and this should be highlighted in the discussion.

Point-by-point reply to Reviewers:

Reviewer #1 (Remarks to the Author):

The authors performed numerous experiments to address concerns raised by both reviewers, and manuscript has been extensively revised. Most of my concerns have been addressed -- through a combination of new experimental data and text editing (where some of the sweeping conclusions were toned down). One remaining concern needs to be addressed. A major conclusion of the paper is summarized in the Introduction (lines 121-123):

“Hyperpolyploid CL hepatocytes are an oncogenerative source of preneoplastic cells through uncharacterized genome reduction processes.”

I agree this is the mostly likely conclusion from the data. However, despite abundant supporting data, there is no direct evident that hyperpolyploid CL hepatocytes are the source. I think it is important to revise the sentence in a manner similar to this: “CL hepatocytes, which are highly enriched with hyperpolyploid cells, are the source of preneoplastic cells, likely through uncharacterized genome reduction processes.” The authors should also edit any other relevant sections/statements accordingly.

Reply:

We thank the reviewer for the good words and we agree. We accepted reviewer’s comment and have toned down our description as follow.

(main text)

“The CL hepatocytes, characterized by their ability to become hyperpolyploid, are the source of preneoplastic cells, likely through uncharacterized genome reduction processes.” (line 121-123)

“Together these results show that hyperpolyploid-enriched CL hepatocytes may have high tumorigenicity and are likely candidates for the origin of the small preneoplastic cells seen after DEN treatment.” (line 285-288)

“In comparison with the smaller size PL-enriched hepatocytes, CL-enriched hepatocytes indeed display higher potential to generate spheres and tumor cells, although it is not possible to exclude that some other low-abundance cells included in the CL fraction during isolation may also have the opportunity to give rise to tumor cells.” (line 476-480)

“Although we did not confirm whether a genome reduction actually occurred, we

observed multipolar dividing in CL hepatocytes-derived tumor cells, which may be an indication of a genome reduction event (Fig. 3k)” (line 496-499)

Reviewer #2 (Remarks to the Author):

I am grateful for the additional work provided by the authors in the revision of the manuscript. The additional transplantation experiments are particularly welcome and are a significant addition. Overall these revisions have improved the manuscript significantly. In response to my comments.

1-1. The additional of the transplant experiment as suggested by myself and reviewer #1 is a helpful addition. This still fails to show direct evidence that the large cells themselves go onto form tumors in the transplant model as this is still a mixed population which is injected. It is crucial that it is explicitly stated that the spheres injected in these assays are appropriately controlled. “Spheres derived from 2000 numbers of hepatocytes were used for implantation.” This implies equal hepatocytes were plated and any resulting spheres were injected. There needs to be appropriate controls injecting equal numbers of spheres from each condition to show that some spheres are more aggressive depending on from where they were obtained.

Reply:

We appreciate this reviewer’s suggestion, which is what we had planned to do in the beginning. However, when seeding the same amount of hepatocytes, the number and size of tumorspheres derived from CL hepatocytes is different from those obtained with PL hepatocytes. Given that the size of tumorspheres can vary from 100 μ m to 170 μ m (Supplementary Fig. 3f), implanting the same number of tumorspheres of the same exact dimensions from all conditions may at least partly cancel the difference between CL and PL hepatocytes, *i.e.* the ability of CL hepatocytes to form more and bigger tumorspheres. Injecting the same amount of tumorspheres without controlling for size would mean injecting different number of cells, due to the difference in size between CL and PL tumorspheres. Therefore, we judged that the same starting seeding density was a more reasonable and technically feasible point of control.

Notably, sphere-forming system is commonly used to retrospectively confirm a certain population of cells with tumorous stem characteristics [1-3]. Here, the higher tumorigenicity of CL hepatocytes after DEN treatment has been demonstrated, which expresses higher numbers and larger size of colony and tumorsphere than those of PL

hepatocytes (Fig. 3g and Supplementary Fig. 3d-f). The numbers and size of tumorspheres obviously reflects the proliferation of sphere forming cells. Together, our previous results have suggested that CL-enriched hepatocytes after DEN-treatment indeed display higher potential to generate tumor cells.

1-2. As there are still smaller hepatocytes in the extracts from CL it is still possible that these are actual origin of the tumors. The evidence presented associatively links the two and the demonstration of frequent multipolar mitosis is supportive, however the possibility still exist that the smaller not the larger cells are those which give rise to foci. Whether these smaller cells' predecessors were ever larger remains to be seen. This could be discussed more clearly and the current wording revised:

“Although we did not formally investigate the mechanisms underlying genomic reduction”: Neither the mechanisms nor an absolute link were proven in this report.

“Together these results show that hyperpolyploid hepatocytes have high tumorigenicity and may be the origin of the small preneoplastic cells seen after DEN treatment.” There was still no direct link between the hyperpolyploid hepatocytes themselves and the foci.

“In conclusion, here we revealed that, after DEN-induced carcinogenic liver injury, hyperpolyploid hepatocytes arising from abscission failure are oncogenerative cells and a major source of preneoplastic lesions.” should be revised to "In conclusion, our data strongly suggests that, after DEN-induced carcinogenic liver injury, hyperpolyploid hepatocytes arising from abscission failure are oncogenerative cells and a major source of preneoplastic lesions.”

Reply:

We thank the reviewer for this suggestion. We have added new sentences in the DISCUSSION and revised inappropriate words as follows.

(main text)

“In comparison with the smaller size PL-enriched hepatocytes, CL-enriched hepatocytes indeed display higher potential to generate spheres and tumor cells, although it is not possible to exclude that some other low-abundance cells included in the CL fraction during isolation may also have the opportunity to give rise to tumor cells.” (line 476-480)

“Although we did not confirm whether a genome reduction actually occurred, we

observed multipolar dividing in CL hepatocytes-derived tumor cells, which may be an indication of a genome reduction event (Fig. 3k).” (line 496-499)

“Together these results show that hyperpolyploid-enriched CL hepatocytes may have high tumorigenicity and are likely candidates for the origin of the small preneoplastic cells seen after DEN treatment.” (line 285-288)

“In conclusion, our data strongly suggests that, after DEN-induced carcinogenic liver injury, hyperpolyploid hepatocytes arising from abscission failure are oncogenerative cells and a major source of preneoplastic lesions.” (line 508-510)

1-3. Consistency re AZD injection. The manuscript states 1 week after DEN but the methods state 1 day after DEN.

Reply:

We thank reviewer for pointing this typo out. AZD was injected at one week after DEN treatment, and we have corrected it now.

(Supplementary material)

To suppress AURKB activity *in vivo*, AZD1152 (25 or 50 mg/kg body weight) was injected one week after DEN injection through i.p. every two days for DEN+AZD group. (line 95)

1-4. Similarly the long term effects of early AZD1152 treatment on carcinogenesis are welcome and a helpful addition to the revised manuscript. I would recommend this important data on the prevention of the cancer phenotype is incorporated in the main Figures.

Reply:

We thank reviewer's suggestion, we have reorganized Supplementary Fig. 7f and g as Fig. 7h and i.

**2. The additional models, again suggested by Reviewer #1 also, are helpful. However, CCl4 is an agent which likes DEN causes pericentral injury and DNA damage. NAFLD is a disease which again is predominantly pericentral in fat loading. Aflatoxin B1 additionally requires activation by specific CYPs which are also zoned. Hence in the models shown it is unclear whether the hyperpolyploidisation is a feature of injury of early carcinogenesis. This should concern should be discussed
This new data Fig S1k should be quantified.**

Reply:

We thank the reviewer for these suggestions, we have added new sentences in DISCUSSION and the quantified data of Supplementary Fig. 1k as follows.

(main text)

Our study demonstrated that xenobiotics (DEN, AFTB1, and CCl₄) and HFD can similarly induce hyperpolyplodization of CL hepatocytes, which may be caused by pericentral injury^{26,51,52}. These models suggested that insults to CL hepatocytes is a common pathway leading to hyperpolyplodization, but the mechanisms underlying early carcinogenesis in these different models may nevertheless be different and remain to be investigated. (line 458-463)

(Supplementary material)

Quantitative data of Supplementary Fig. 1k

3. I accept the author's comments that this is out with the scope of the current manuscript. The additional details are welcome to address the DEN timepoint series and the methodology.

Reply:

We thank the reviewer for this viewpoint and we agree.

4-1. (Additional points) The lack of significance appears to be a type 2 error due to underpowering of the experiments. Now that it is apparent that neither of these key results reach a reasonable cut off for significance. I would suggest increasing the replicates to test whether there is a genuine and demonstrable effect on complete cytokinesis and abscission failure.

Reply:

We agree with this analysis. We performed one additional independent experiment

and added the results to the Fig. 5e. Significant difference between DEN and DEN+AZD groups was reached in both complete cytokinesis and abscission failure. New Fig. 5e is displayed as follows.

e

4-2. I agree with the changes made in response to my minor concerns.

Reply:

We are grateful to the reviewer for recognizing our works.

4-3. Further proof reading and corrections are required e.g. Line 176 “The understand the cause of”, and the supplemental information.

Reply:

Thank you for noticing this. We have corrected inappropriate sentences in main text as follows.

(main text)

To understand the cause of hyperpolyploidy, we next investigated whether cell proliferation or cytokinesis was affected by DEN. (line 176)

4-4. I agree with the comments from reviewer #1, which are broadly in line with my own. I would, in response to the authors reply, highlight that the additional markers of laminB1 and gh2Ax are helpful. However, I would caution the use of “senescence” without direct evidence of lack of proliferation in these cells and encourage the authors not to imply that cells in this area, which is the most proliferative are senescence. LaminB1 is associated with senescence but is not a bona fide marker of senescence – few are, as the authors will be aware.

Reply:

This is a matter of debate. Quite a few papers by leaders in the field (notably Judith

Campisi, Oliver Dreesen, and Robert D. Goldman) have proposed Lamin B1 as a marker of senescence [4-6]. Nonetheless, we have modified the main text to satisfy this suggestion.

(main text)

In contrast with γ H2AX, downregulation of Lamin B1 was observed in DEN-treated liver, which was specifically identified in hepatocytes but not in other cell types such as portal triad endothelial cells (yellow arrow). ~~Importantly, lowest Lamin B1 expression was seen in preneoplastic cells, compared to adjacent hepatocytes~~ (Supplementary Fig. 2e,f).
(line 215-218)

4-5. The delineation of preneoplastic and neoplastic is still arbitrary and this should be highlighted in the discussion.

Reply:

We apologize for not including references about the definition for preneoplastic lesions. The sequentially pathological changes have been well described in human and rodents, and the criteria of preneoplastic lesions is based on Bannasch's and Su's definition published in 1996 and 2003 [7-8]. It is arbitrary, but the criteria are nonetheless established. We have added sentences and uploaded the references in main text as follows.

(main text)

Because hyperpolyploid hepatocytes with precancerous traits were mainly located within the CL and ML regions, we next sought to determine whether preneoplastic lesions, ~~a small cell change of the liver parenchyma with microscopic foci of altered hepatocyte~~ (see definition in Supplementary material)^{31,32}, also predominantly occurred in these regions.
(line 220-221)

Microscopic foci of preneoplastic lesions, ~~based on previous definition~~^{31,32}, adjacent the CV marker were observed in DEN-injected liver, surrounded by hyperpolyploid hepatocytes (Fig. 1 and 2)...(line 450)

References

1. Yamashita, T., et al., *EpCAM-positive hepatocellular carcinoma cells are tumor-initiating cells with stem/progenitor cell features*. Gastroenterology, 2009. **136**(3): p. 1012-24.
2. Lee, T.K., et al., *CD24(+) liver tumor-initiating cells drive self-renewal and tumor initiation through STAT3-mediated NANOG regulation*. Cell Stem Cell, 2011. **9**(1): p. 50-63.
3. Yang, W., et al., *OV6(+) tumor-initiating cells contribute to tumor progression and invasion in human hepatocellular carcinoma*. J Hepatol, 2012. **57**(3): p. 613-20.
4. Freund, A., et al., *Lamin B1 loss is a senescence-associated biomarker*. Mol Biol Cell, 2012. **23**(11): p. 2066-75.
5. Dreesen, O., et al., *Lamin B1 fluctuations have differential effects on cellular proliferation and senescence*. J Cell Biol, 2013. **200**(5): p. 605-17.
6. Shimi, T., et al., *The role of nuclear lamin B1 in cell proliferation and senescence*. Genes Dev, 2011. **25**(24): p. 2579-93.
7. Bannasch, P., *Pathogenesis of hepatocellular carcinoma: sequential cellular, molecular, and metabolic changes*. Prog Liver Dis, 1996. **14**: p. 161-97.
8. Su, Q. and P. Bannasch, *Relevance of hepatic preneoplasia for human hepatocarcinogenesis*. Toxicol Pathol, 2003. **31**(1): p. 126-33.

Table 1. List of revised Figures

Items	Event	Revised
1.	Make rearrangement of Fig. 7 and Supplementary Fig. 7	According to Reviewer #2's comment, we moved Supplementary Fig. 7f and g to main Fig. 7h and I, respectively.
2.	Add new figures in Supplementary Fig. 1	According to Reviewer #2's comment, we added new quantitative figures as Supplementary Fig. 1k.
3.	Typos correction	We corrected the typo of area unit from " μm " to " μm^2 " in Fig. 1d, Fig. 2b, Fig. 7d, and Supplementary Fig. 1c, f, g, and Fig. 7c.

Table 2. List of the critical revise of manuscript

Items	Event	Revised
1.	Tone down description in INTRODUCTION (line 121-123)	The CL hepatocytes, characterized by their ability to become hyperpolyploid, are the source of preneoplastic cells, likely through uncharacterized genome reduction processes.
2.	Correct typo in RESULTS (line 176)	To understand the cause of hyperpolyploidy, we next investigated whether cell proliferation or cytokinesis was affected by DEN.
3.	Add new words and references in RESULTS (line 219-222)	...we next sought to determine whether preneoplastic lesions, a small cell change of the liver parenchyma with microscopic foci of altered hepatocytes (see definition in Supplementary material) ^{31,32} , also predominantly occurred in these regions.
4.	Tone down description in RESULTS (line 285-288)	Together these results show that hyperpolyploid-enriched CL hepatocytes may have high tumorigenicity and are likely candidates for the origin of the small preneoplastic cells seen after DEN treatment.
5.	Replace Supplementary Fig. 7f,g by Fig. 7h,i in RESULTS	Moreover, AZD1152 treatment also partially reduced the size of the larger tumor nodules observed on the surface of the liver 6 months after DEN treatment (Supplementary Fig. 7f,g Fig. 7h,i).
6.	Add new words and references in DISCUSSION (line 450-451)	Microscopic foci of preneoplastic lesions, based on previous definition ^{31,32} , adjacent the CV marker were observed in DEN-injected liver,...
7.	Add new sentences in DISCUSSION (line 458-465)	Our study demonstrated that xenobiotics (DEN, AFTB1, and CCl4) and HFD can similarly induce hyperpolyploidization of CL hepatocytes, which may be caused by pericentral injury ^{26,51,52} . These models suggested that insults to CL hepatocytes is a common

		pathway leading to hyperpolyploidization, but the mechanisms underlying early carcinogenesis in these different models may nevertheless be different and remain to be investigated. Most importantly, our results reveal a molecular correlation between pathological hyperpolyploidy of CL hepatocytes and DEN-induced HCC formation in mouse.
8.	Add new sentences in DISCUSSION (line 476-480)	In comparison with the smaller size PL-enriched hepatocytes, CL-enriched hepatocytes indeed display higher potential to generate spheres and tumor cells, although it is not possible to exclude that some other low-abundance cells included in the CL fraction during isolation may also have the opportunity to give rise to tumor cells.
9.	Tone down description in DISCUSSION (line 496-499)	Although we did not confirm whether a genome reduction actually occurred, we observed multipolar dividing in CL hepatocytes-derived tumor cells, which may be an indication of a genome reduction event (Fig. 3k).
10.	Tone down description in DISCUSSION (line 508)	In conclusion, our data strongly suggests that, after DEN-induced carcinogenic liver injury, ...
11.	Revise experimental numbers in the legend of Fig. 5e (line 846)	(e) Quantification of the abscission failure ratio in the indicated cultured hepatocytes from time-lapse images. n > 500 dividing hepatocytes/group, four independent experiments/group.
12.	Add new sentences in the legend of Fig. 7h and i (line 899-904)	(h) Representative livers from control (Ctrl), DEN-treatment (DEN), DEN-treatment followed by two months AZD1152 (DEN+AZD) injection were sampled six months after DEN treatment for tumor size and numbers examination. (i) Quantitative data show the tumor diameter and tumor numbers in Fig. 7f. N = 5 mice/group. Statistic: One-way ANOVA, (b), (d), (f), (g)

		and (i); Two-way ANOVA, (e). Values represent the mean \pm s.e.m., * P < 0.05, ** P < 0.01, *** P < 0.001. n.s., not significant; n.d., not detectable. Scale bars: 10 mm in (h).
13.	Correct typo in Supplementary Material (line 95)	To suppress AURKB activity in vivo , AZD1152 (25 or 50 mg/kg body weight) was injected one week after DEN injection through i.p. every two days for one or two months.
14.	Add new references in Supplementary Material (line 210)	The sequentially pathological changes have been well described in human and rodents, and the terminology and criteria for various lesions has been defined based on consensus diagnosis ^{31,32} .
15.	Add new words in Supplementary Material (line 358 and 368)	...For the effect of time on nuclear, cellular and micronuclear size after drugs treatment (Fig 1d, Fig. 6b, Fig. 7d, Supplementary Fig. 1f and k,and the tumor diameter and tumor numbers in Fig. 7f, g, and i were statistically analyzed by One-way ANOVA.
16.	Add new words in the legend of Supplementary Fig. 1k (line 456-458)	The nucleus and cell size were analyzed and showed as dot plot graph. Mouse number (N) = 3 per group, and cell number (n) = 300 per group. One-way ANOVA with Bonferroni's post-test was used to (c), (f), and (k);...

REVIEWERS' COMMENTS

Reviewer #2 (Remarks to the Author):

Responses to my comments

1.1 I accepted the revised wording of the discussion which now does not overstate the interpretation of the sphere transplantation experiment.

I still have significant concern given the authors' clarification about this this experiment was controlled. Taking 2000 cells, seeding them and growing spheres before transplanting those resultant spheres, is not an assay to test the potency of the spheres themselves unless the spheres are equivalent at the time of implantation. The assay they describe is likely producing results based as much on the ability to form spheres in vitro from the different zones as the ability of those spheres to form tumors in vivo. Ideally this experiment should be repeated with equivalent injections.

1.2 I accept the revised wording made by the authors.

1.3 Correction appreciated.

1.4 Reorganisation appreciated.

2 Clarification in the discussion is appreciated.

3 No further action required

4 The additional replicate is helpful to now show significance.

4.2 No further action required

4.3 No further action required at this stage

4.4 I appreciate the changes to the text as recommended.

4.5 The clarification provided is appreciated

Point-by-point reply to Reviewer#2:

1.1 I accepted the revised wording of the discussion which now does not overstate the interpretation of the sphere transplantation experiment. I still have significant concern given the authors clarification about this this experiment was controlled. Taking 2000 cells, seeding them and growing spheres before transplanting those resultant spheres, is not an assay to test the potency of the spheres themselves unless the spheres are equivalent at the time of implantation. The assay they describe I likely producing results based as much on the ability to form spheres in vitro from the different zones as the ability of those spheres to form tumors in vivo. Ideally this experiment should be repeated with equivalent injections.

Reply:

Thanks for reviewer's suggestion and we agree with this point; however, the heterogeneous size of spheres makes it difficult to obtain the same starting seeding density. Previous study demonstrated that 200 and 1000 primary hepatocellular carcinoma sphere cells initiate tumors in 3 of 6 injected mice after 6 week of subcutaneous transplantation,¹ suggesting intrinsic transformation activity contributed more than cell number on spheres-mediated tumor formation. Here, we subcutaneously implanted of spheres derived from CL and PL hepatocytes into the right and left flanks of NSG immunodeficient mice. CL hepatocyte-derived spheres could initiate tumors in 4 of 6 injected mice, whereas PL hepatocyte-derived spheres produced two very small size clumps among 6 injected mice after three months of transplantation. Around 200 PL hepatocyte sphere cells had been injected into mice, while no tumor formation in PL hepatocyte-injected mice. Although PL hepatocyte may initiate tumor formation at an increased number of sphere cells, CL hepatocyte-derived spheres indeed display highest tumorigenic capacity than PL hepatocyte-derived spheres following the present protocol. We therefore provide a suggestion that CL hepatocytes after DEN-treatment display higher potential to generate tumor cells.

Reference

1. Ma XL, Sun YF, Wang BL, et al. Sphere-forming culture enriches liver cancer stem cells and reveals Stearoyl-CoA desaturase 1 as a potential therapeutic target. *BMC Cancer*. 2019;19(1):760.

- 1.2 I accept the revised wording made by the authors.*
- 1.3 Correction appreciated.*
- 1.4 Reorganisation appreciated.*
- 2 Clarification in the discussion is appreciated.*
- 3 No further action required*
- 4 The additional replicate is helpful to now show significance.*
- 4.2 No further action required*
- 4.3 No further action required at this stage*
- 4.4 I appreciate the changes to the text as recommended.*
- 4.5 The clarification provided is appreciated*

Reply:

We appreciate the Reviewer#2 for acknowledging the importance of our work.